# HOLISTIC ADVERSARIALLY ROBUST PRUNING

**Qi Zhao and Christian Wressnegger**
KASTEL Security Research Labs, Karlsruhe Institute of Technology, Germany

## ABSTRACT

Neural networks can be drastically shrunk in size by removing redundant parameters. While crucial for the deployment on resource-constraint hardware, oftentimes, compression comes with a severe drop in accuracy and lack of adversarial robustness. Despite recent advances, counteracting both aspects has only succeeded for moderate compression rates so far. We propose a novel method, HARP, that copes with aggressive pruning significantly better than prior work. For this, we consider the network holistically. We learn a global compression strategy that optimizes how many parameters (compression rate) and which parameters (scoring connections) to prune *specific to each layer individually*. Our method fine-tunes an existing model with dynamic regularization, that follows a step-wise incremental function balancing the different objectives. It starts by favoring robustness before shifting focus on reaching the target compression rate and only then handles the objectives equally. The learned compression strategies allow us to maintain the pre-trained model's natural accuracy and its adversarial robustness for a reduction by $99\,\%$ of the network's original size. Moreover, we observe a crucial influence of non-uniform compression across layers. The implementation of HARP is publicly available at `https://intellisec.de/research/harp`.

## 1 INTRODUCTION

Deep neural networks (DNNs) yield remarkable performances in classification tasks in various domains (He et al., 2016; Cakir & Dogdu, 2018; Schroff et al., 2015; Li et al., 2022) but are vulnerable to input manipulation attacks such as adversarial examples (Szegedy et al., 2014). Small perturbations to benign inputs can cause worst-case errors in prediction. To date, the most promising defensive approach against this sort of attacks is adversarial training as introduced by Madry et al. (2018) and further refined ever since (Shafahi et al., 2019; Zhang et al., 2019; Wang et al., 2020). It introduces adversarial examples into the training process, diminishing the generalization gap between natural performance and adversarial robustness. However, there is evidence indicating that higher robustness requires over-parameterized networks that have wider layers and higher structural complexity (Madry et al., 2018; Zhang et al., 2019; Wu et al., 2021), rendering the task of combining both objectives—compression and robustness—inherently difficult.

Neural network pruning (Han et al., 2016; Yang et al., 2017; He et al., 2017), for instance, has been proven to be an extraordinary valuable tool for compressing neural networks. The model can be reduced to a fraction of its size by removing redundancy at different structural granularity (Li et al., 2017; Mao et al., 2017; Kollek et al., 2021). However, pruning inflicts a certain recession in model accuracy (Han et al., 2016) and adversarial robustness (Timpl et al., 2021) that is unavoidable the stronger the model is compressed. The aim of adversarial robust pruning hence is to maintain the accuracy and robustness of an already adversarially pre-trained model as good as possible. Despite great efforts (Ye et al., 2019; Sehwag et al., 2020; Madaan et al., 2020; Özdenizci & Legenstein, 2021; Lee et al., 2022), this dual objective has only been achieved for moderate compression so far.

In this paper, we start from the hypothesis that effective adversarially robust pruning requires a non-uniform compression strategy with learnable pruning masks, and propose our method HARP. We follow the three-stage pruning pipeline proposed by Han et al. (2015) to improve upon pre-trained models, where we jointly optimize score-based pruning masks and layer-wise compression rates during fine-tuning. As high robustness challenges the compactness objective (Timpl et al., 2021), we employ a step-wise increasing weighted-control of the number of weights to be pruned, such that we can learn masks and rates simultaneously. Our approach explores a global pruning strategy that allows for on-par natural accuracy with little robustness degradation only.

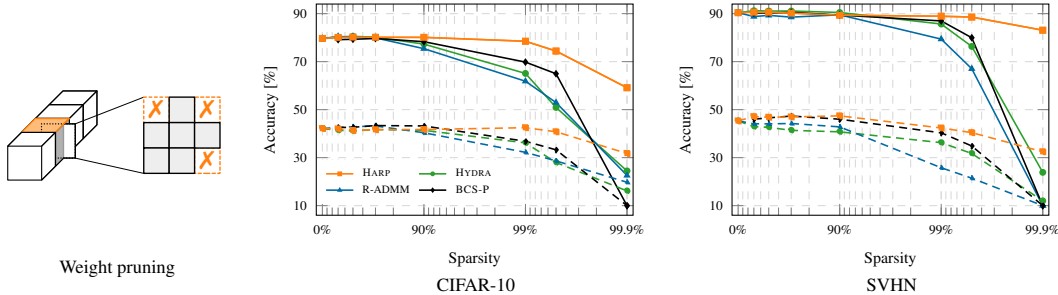

Figure 1: Overview of pruning weights of a VGG16 model for CIFAR-10 (left) and SVHN (right) with PGD-10 adversarial training. Solid lines show the natural accuracy of HARP, HYDRA, Robust-ADMM and BCS-P (cf. Table 7). Dashed lines represent the robustness against AUTOATTACK.

In summary, we make the following contributions:

- **Novel pruning technique for pre-trained models.** We optimize how many parameters and which parameters to prune *for each layer individually*, resulting in a global but non-uniform pruning strategy. That is, the overall network is reduced by a predetermined rate governed by the target hardware's limits, but layers are compressed varyingly strongly. We show that both aspects are needed for HARP to take full effect (cf. Section 4.1).

- **Significant improvement over related work.** An overview of our method's performance is presented in Fig. 1 for a PGD-AT trained VGG16 model learned on CIFAR-10 (left) and SVHN (right), providing a first glimpse of the yield advances. No less importantly, we conduct experiments with small (CIFAR-10, SVHN) as well as large-scale (ImageNet) datasets across different robust training methods and various attacks (cf. Section 4.2).

- **Importance of non-uniform pruning for adversarial robustness.** We emphasize the superiority of non-uniform strategies (Zhao et al., 2022) by extending existing adversarially robust pruning-techniques. We demonstrate that HYDRA (Sehwag et al., 2020) and Robust-ADMM (Ye et al., 2019) yield better results when used with non-uniform strategies determined by ERK (Evci et al., 2020) and LAMP (Lee et al., 2021) (cf. Section 4.3).

## 2 NEURAL NETWORK PRUNING

Removing redundant parameters reduces a network's overall memory footprint and necessary computations, allowing for a demand-oriented adaptation of neural networks to resource-constrained environments (Han et al., 2015; 2016; Wen et al., 2016; Huang et al., 2018; He et al., 2018; 2017; Li et al., 2017; Mao et al., 2017; Molchanov et al., 2017). Neural network pruning attempts to find a binary mask $M^{(l)}$ for each layer $l$ of a network with $L$ layers represented by its parameters $\theta$ and $\theta^{(l)}$, respectively. The overall pruning-mask thus is denoted as $M = (M^{(1)}, \ldots, M^{(l)}, \ldots, M^{(L)})$. These masks specify which parameters of the layer $\theta^{(l)}$ to remove (zero out) and which to keep, yielding a reduced parameter set $\hat{\theta}^{(l)} = \theta^{(l)} \odot M^{(l)}$, where $\odot$ is the Hadamard product. Based on this, we define the overall compression rate $a$ as the ratio of parameters preserved after pruning, $\Theta_{\neq 0}$, to the number of parameters in the model, $\Theta$. Compression rates for individual layers $a^{(l)}$ are defined analogously. Note, that a network's *sparsity* is defined inversely, meaning, a 99.9 % sparsity refers to a compression of 0.001. In further follow, we consider $\theta^{(l)} \in \mathbb{R}^{c_i^{(l)} \times c_o^{(l)} \times k^{(l)} \times k^{(l)}}$ where $c_i^{(l)}$ and $c_o^{(l)}$ represent input and output channels, respectively, and $k^{(l)}$ is the kernel size.

Han et al. (2015) propose a three stage pipeline for network pruning, starting with (1) training an over-parameterized network, followed by (2) removing redundancy per layer according to some pruning criterion, before (3) recovering network performance via fine-tuning. The actual pruning strategy, that decides on the pruning mask $M$, is obtained in the second step. While integrated approaches (pruning during model training) exist (e.g., Vemparala et al., 2021; Özdenizci & Legenstein, 2021), the staged process remains most common (Liu et al., 2019; Sehwag et al., 2020; Lee et al., 2022) as it allows to benefit from recent advantages in adversarial training (Shafahi et al., 2019; Zhang et al., 2019; Wang et al., 2020) out-of-the-box.

## 3 HOLISTIC ADVERSARIALLY ROBUST PRUNING

Two aspects are crucial when pruning neural networks: *How many and which parameters to prune.* Current state-of-the-art methods for adversarially robust pruning focus on the latter, allocating a fixed compression budget governed by the target hardware's limits and using it uniformly for each layer (e.g., Ye et al., 2019; Sehwag et al., 2020). We argue that learning the optimal pruning amount—the compression rate per layer—is equally crucial. HARP thus integrates measures for compression rates and scoring connections in an empirical min-max optimization problem based on adversarial training to learn a suitable pruning mask $M$ for a pre-trained model $\boldsymbol{\theta}$:

$$\min_{\boldsymbol{r},\boldsymbol{S}} \quad \mathbb{E}_{(\boldsymbol{x},y)\sim\mathcal{D}} \left[ \max_{\delta} \left\{ \mathcal{L}_{robust}(\boldsymbol{\theta} \odot \boldsymbol{M}, \boldsymbol{x} + \delta, y) \right\} \right] + \gamma \cdot \mathcal{L}_{hw}(\boldsymbol{\theta} \odot \boldsymbol{M}, a_t), \tag{1}$$

The inner maximization generates adversarial examples from a benign input $\boldsymbol{x}$ with label $y$ from a distribution $\mathcal{D} = \{\boldsymbol{x}, y\}$ by adding noise $\delta$, and incorporates these when minimizing the training loss $\mathcal{L}_{ce}$ (Madry et al., 2018). The exact formulation of $\mathcal{L}_{robust}$ depends on the pre-trained model, for which we evaluate PGD-AT (Madry et al., 2018), TRADES-AT (Zhang et al., 2019), and MART-AT (Wang et al., 2020). Simultaneously, the model is subject to a pruning mask $M$ that involves two parameters: First, the compression quota $\boldsymbol{r}$, which is a learnable representation of compression rates, and second, scores $\boldsymbol{S}$ for determining the importance of the network's connections. In the following, we discuss HARP's global compression control and describe how pruning strategies are learned, before we elaborate on how to dynamically regularize the training process using $\gamma$ by following a step-wise incremental function balancing the different objectives.

**Global compression control.** The compression control is encoded as an additional loss term $\mathcal{L}_{hw}$, that considers compression for the entire network to reach a specific target compression $a_t$. However, it explicitly allows for layer-specific rates, yielding an overall non-uniform compression strategy: We measure the number of currently preserved (i.e., non-zero) weights across all $L$ layers of the network, $\Theta_{\neq 0} = \sum_{l=1}^{L} \| (\mathbb{1}_{w\neq 0})_{w\in\hat{\boldsymbol{\theta}}^{(l)}} \|_1$, relative to the targeted total number of parameters at a compression rate $a_t$:

$$\mathcal{L}_{hw}(\hat{\boldsymbol{\theta}}, a_t) := \max \left\{ \frac{\Theta_{\neq 0}}{a_t \cdot \Theta} - 1 \, , \, 0 \right\} \tag{2}$$

Additionally, we clip rates that would exceed the targeted compression-rate. Constraining the loss to positive values ensures to not encourage even lower compression rates that would potentially harm accuracy and robustness unnecessarily.

For learning the layer-specific compression rates, moreover, it is crucial that we constrain the reduction by a minimal compression $a_{min} = 0.1 \times a_t$ as we need to prevent a layer from been removed completely. We thus limit the layers' compression $a^{(l)}$ to $[a_{min}, 1]$. In order to better control learning of the optimal pruning rate in this range, we introduce a trainable parameter $r^{(l)} \in \mathbb{R}$, which we refer to as *compression quota*, and constrain it by an activation function $g : \mathbb{R} \to [a_{min}, 1]$. For the sake of the continuous derivability on $r$, we use the sigmoid function squeezed to the desired output range, yielding a layer-wise compression rate $a^{(l)} = g(r^{(l)})$, where $g : r \mapsto (1 - a_{min}) \cdot \text{sig}(r) + a_{min}$.

**Connection importance.** For learning the actual pruning mask, in turn, we introduce another learnable parameter $\boldsymbol{S}$, the connection importance-score matrix, to rate each pruning connection. In weight pruning, $\boldsymbol{S}^{(l)}$ originates $\mathbb{R}^{c_i^{(l)} \times c_o^{(l)} \times k^{(l)} \times k^{(l)}}$, and is assigned values from activation function $g$. Additionally, we use $P(\alpha^{(l)}, \boldsymbol{S}^{(l)})$ to represent the function that determines the $\alpha$-th percentile on layer $l$ as the cutoff threshold for pruning parameters in accordance with compression rate $a^{(l)}$, that is: $\alpha^{(l)} = 1 - a^{(l)}$. Due to the relation with $\boldsymbol{S}^{(l)}$ (and therefore also with $r^{(l)}$ via compression rate $a^{(l)}$), the binary pruning mask $\boldsymbol{M}^{(l)}$ is thus trainable and defined as:

$$\boldsymbol{M}^{(l)} := \left( \mathbb{1}_{s > P(\alpha^{(l)}, \boldsymbol{S}^{(l)})} \right) \tag{3}$$

Pruning the $l$-th layer can hence be expressed by the Hadamard product of the mask and the model's parameters $\boldsymbol{\theta}$ yielding the pruned model parameters $\hat{\boldsymbol{\theta}}^{(l)} = \boldsymbol{\theta}^{(l)} \odot \boldsymbol{M}^{(l)}$.

**Learning pruning strategies.**    Binarizing importance scores to obtain a pruning mask as described above is a non-differentiable operation and prevents us from performing gradient descent over the masks. We, thus, follow the "straight through estimation" (STE) strategy (Hubara et al., 2016) as proposed by Kusupati et al. (2020) to assign the updated gradients to the importance scores $\boldsymbol{S}^{(l)}$ directly and proceed similarly for compression quotas $r^{(l)}$:

$$\frac{\partial \mathcal{L}}{\partial \boldsymbol{S}^{(l)}} = \frac{\partial \mathcal{L}}{\partial \hat{\boldsymbol{\theta}}^{(l)}} \cdot \frac{\partial \hat{\boldsymbol{\theta}}^{(l)}}{\partial \boldsymbol{M}^{(l)}} \cdot \frac{\partial \boldsymbol{M}^{(l)}}{\partial \boldsymbol{S}^{(l)}} \qquad \overset{\textbf{STE!}}{=} \frac{\partial \mathcal{L}}{\partial \hat{\boldsymbol{\theta}}^{(l)}} \cdot \frac{\partial \hat{\boldsymbol{\theta}}^{(l)}}{\partial \boldsymbol{M}^{(l)}} \tag{4}$$

$$\frac{\partial \mathcal{L}}{\partial r^{(l)}} = \frac{\partial \mathcal{L}}{\partial \hat{\boldsymbol{\theta}}^{(l)}} \cdot \frac{\partial \hat{\boldsymbol{\theta}}^{(l)}}{\partial \boldsymbol{M}^{(l)}} \cdot \frac{\partial \boldsymbol{M}^{(l)}}{\partial g(r^{(l)})} \cdot g'(r^{(l)}) \overset{\textbf{STE!}}{=} \langle \frac{\partial \mathcal{L}}{\partial \hat{\boldsymbol{\theta}}^{(l)}} \cdot \frac{\partial \hat{\boldsymbol{\theta}}^{(l)}}{\partial \boldsymbol{M}^{(l)}} \rangle \cdot g'(r^{(l)}) \tag{5}$$

Finally, also the initialization of the connection importance-scores is crucial. We follow the lead of Sehwag et al. (2020) and initialize the scores proportional to the weights of the pre-trained model and use a scaling factor $\eta = \sqrt{k/\textit{fan-in}^{(l)}}$ with $k = 6$ and *fan-in* being the product of the receptive field size and the number of input channels. Our initialization for pruning weights, hence, follows the order of weight magnitudes (Ye et al., 2019):

$$\boldsymbol{S}_W^{(l)} = \left( \eta \cdot \frac{\boldsymbol{\theta}^{(l)}}{\max(|\boldsymbol{\theta}^{(l)}|)} \right)_{c_i^{(l)} \times c_o^{(l)} \times k^{(l)} \times k^{(l)}} \tag{6}$$

HARP does not use a fixed ratio for pruning parameters, but learns the optimal compression rate per layer. However, for initializing the compression quota $r$, we first align with a uniform compression strategy that sets the same rate $a_{init}$ on each layer and let the optimizer improve it (cf. Appendix A.3 for more details). For the activation function $g(\cdot)$ defined above, we thus yield:

$$r^{(l)} = \log \left( \frac{a_{init} - a_{min}}{1 - a_{init}} \right) \tag{7}$$

**Balancing pruning objectives.**    HARP strives to strike a balance between natural accuracy, adversarial robustness and compactness. For this, we employ a step-wise incremental function to adapt the regularization parameter $\gamma$ throughout the learning process. We start by favoring robustness before shifting focus on reaching the target compression rate and only then handle the objectives equally.

Fig. 2 shows the relation of $\gamma$ and the two different losses, $\mathcal{L}_{robust}$ against PGD-10 and $\mathcal{L}_{hw}$, on the y-axis over the epochs on the x-axis, exemplarily for weight pruning VGG16 (learned on CIFAR-10) with a target compression $a_t = 0.01$, that is, removing 99 % of the parameters. The model starts off from the uniform strategy with $a_{init} = 0.1$, learning to be more robust while being only slightly penalized by $\mathcal{L}_{hw}$. As the model has been adversarially pre-trained, $\mathcal{L}_{robust}$ is low already. Over time, we put more focus on $\mathcal{L}_{hw}$ up until we have reached our target compression ($\gamma = 0.11$ in this example), causing it to decrease while $\mathcal{L}_{robust}$ increases. Afterwards,

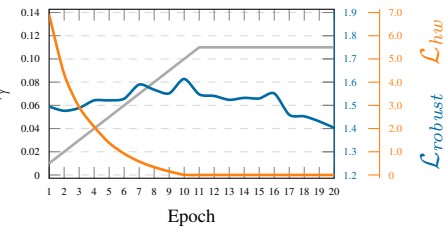

Figure 2: HARP's step-wise regularization of pruning objectives for VGG16 on CIFAR-10 with a target sparsity of 99 %.

the model aims for higher robustness under the found $\mathcal{L}_{hw}$ penalty, balancing both objectives. In Appendix A.1, we study the influence of different step sizes and elaborate on the progression of the model's natural accuracy and adversarial robustness in Appendix A.3.

## 4    EVALUATION

We resume to demonstrate HARP's effectivity in pruning neural networks by comparing to existing state-of-the-art methods in Section 4.2. In Section 4.1, we then perform an ablation study regarding the learnable parameters used by our method. In particular, we show the influence of learning compression quotas $r$ and importance scores $S$, separately, to demonstrate that both are indeed needed for successful, holistic pruning. In Section 4.3 and Appendix A.3, we then discuss the non-uniform nature of HARP's pruning strategy and compare to novel extensions of HYDRA and

Robust-ADMM to non-uniform compression. In the appendix, we additionally present the extended comparison to related work (Appendix A.2), elaborate on the pruned model's parameter distribution (Appendix A.4), show HARP's performance on naturally trained models (Appendix A.5), and extend HARP to channel pruning (Appendix A.6).

**Experimental setup.** We evaluate HARP on two small-scale datasets, CIFAR-10 (Krizhevsky et al.) and SVHN (Netzer et al., 2011). While the first is balanced, the second is not. Consequently, we use accuracy (ACC) as performance measure for the former, and the balanced accuracy (BACC) for the latter (Brodersen et al., 2010). Each dataset is learned with a VGG16 (Simonyan & Zisserman, 2015) and ResNet18 (He et al., 2016) model. Additionally, we show the performance on the large-scale, balanced ImageNet dataset (Deng et al., 2009) trained with a ResNet50 network. We apply $\gamma = 0.01$ for small-scale datasets, and increase $\gamma$ to 0.1 for ImageNet to guarantee the arrival at the target compression rate $a_t$. In the pruning and fine-tuning phase of HARP, we train for 20 epochs and 100 epochs, respectively.

**Adversarial training.** We adversarially train on the small-scale datasets, CIFAR-10 and SVHN, with PGD-AT (Madry et al., 2018), TRADES-AT (Zhang et al., 2019), and MART-AT (Wang et al., 2020). For each, we use $l_\infty$ PGD-10 attacks with random initialization and a perturbation strength $\epsilon = {}^8/_{255}$ with step size $\alpha = {}^2/_{255}$. We adopt stochastic gradient descent with a cosine learning-rate schedule (Loshchilov & Hutter, 2016) and a weight decay of 0.0005 (Pang et al., 2021). For TRADES and MART, we follow the originally proposed regularization of $\lambda = 6.0$ and $\lambda = 5.0$, respectively. To accelerate the training process for the large-scale ImageNet experiments, we resort to FREE-AT (Shafahi et al., 2019) with 4 replays, reducing the pruning and fine-tuning phase to 5 epochs and 25 epochs, and use a perturbation strength $\epsilon = {}^4/_{255}$ with step size $\alpha = {}^1/_{255}$.

**Adversarial robustness evaluation.** To validate the pruned networks' adversarial robustness, we use C&W$_\infty$ (Carlini & Wagner, 2017) optimized by PGD and PGD-10 with the same settings used for training. Furthermore, we implement AUTOPGD (APGD) (Croce & Hein, 2020a) with cross-entropy (CE) loss, 50 steps, and 5 restarts, and AUTOATTACK (AA) (Croce & Hein, 2020a) using the standard ensemble out of AUTOPGD with CE-loss, Targeted-AUTOPGD$_{DLR}$, FAB (Croce & Hein, 2020b), and SQUAREATTACK (Croce et al., 2019). All attacks are carried out on the respective complete test dataset.

## 4.1 ABLATION STUDY

HARP features two important components: Learning compression quotas and learning what connections to prune. While we do argue that both aspects are crucial for effective adversarial robust pruning, in this section, we set out to analyze whether this indeed is the case. We perform an ablation study regarding the compression quota to be learned $r$ and determining importance scores $S$. For the experiments reported in Table 1, we thus either learn compression quotas or importance scores, while keeping the other one constant—consequently, we also do not balance pruning objectives here. Note that for the optimization on compression quotas $r$, importance scores $S$ remain as initialized by Eq. (9), while during investigating the influence of scores $S$, we use uniform compression. We denote our method with compression-rate optimization only as HARP-$r$, and with importance-score optimization only as HARP-$S$.

As already observed in Fig. 1, moderate compression manages to maintain natural accuracy and adversarial robustness very well, such that in this study, we focus on 99 % and 99.9 % sparsity.

Table 1: Optimizing either compression rates (HARP-$r$) or importance scores (HARP-$S$). Natural accuracy and PGD-10 adversarial robustness are presented left and right of the / character.

| Model | Adv. Training | 99 % Sparsity | | | 99.9 % Sparsity | | |
|---|---|---|---|---|---|---|---|
| | | HARP-$r$ | HARP-$S$ | HARP | HARP-$r$ | HARP-$S$ | HARP |
| ResNet18 | PGD | **76.39** / **46.64** | 72.05 / 43.69 | **80.25** / **50.36** | 41.66 / 27.54 | **57.66** / **35.92** | **63.99** / **39.39** |
| | TRADES | 73.31 / 45.14 | **75.50** / **46.37** | **77.78** / **50.16** | 73.31 / 45.14 | **75.50** / **46.37** | **77.78** / **50.16** |
| | MART | 70.08 / **48.38** | **75.27** / 47.11 | **75.88** / **50.79** | 70.08 / **48.38** | **75.27** / 47.11 | **75.88** / **50.79** |
| VGG16 | PGD | **76.17** / **46.74** | 65.09 / 39.80 | **78.50** / **48.71** | 36.76 / 28.02 | **50.33** / **34.03** | **59.13** / **37.36** |
| | TRADES | **72.91** / **44.52** | 66.75 / 41.79 | **76.46** / **48.01** | 41.63 / 26.95 | **56.08** / **31.51** | **63.43** / **34.64** |
| | MART | **71.63** / **48.64** | 64.37 / 41.46 | **73.04** / **51.09** | 37.19 / 30.68 | **49.51** / **36.29** | **55.02** / **39.39** |

Interestingly, HARP-$r$ appears to be particular beneficial for moderate to high compression, while HARP-$S$ is increasingly important for aggressive pruning. In particular for VGG16 with its cascading network architecture, this tendency is apparent for different pre-training strategies. However, at either sparsity level the combination of both, that is, optimizing compression rates $r$ *and* importance scores $S$, allows HARP to excel.

## 4.2 ROBUST PRUNING WITH HARP

In this section, we report results of our comparative evaluation with Robust-ADMM (Ye et al., 2019), HYDRA (Sehwag et al., 2020), BCS-P (Özdenizci & Legenstein, 2021), and MAD (Lee et al., 2022) in pruning a neural network's weights. In particular, for aggressive compression yielding high sparsity, HARP excels and outperforms prior work. For the comparison to Robust-ADMM and HYDRA, we perform a systematic evaluation with CIFAR-10, SVHN, and ImageNet. For BCS-P and MAD, we revert to the respective settings reported in the original papers.

**Comparison to Robust-ADMM.** The method by Ye et al. (2019) is designed to robustly prune neural networks on fine-grained (weights) and coarse-grained granularity (channels/filters). We focus on pruning weights at this point and present further results on pruning channels in Appendix A.6. Tables 2 to 4 summarize the results for pruning models on CIFAR-10, SVHN, and ImageNet, respectively. Robust-ADMM performs well for pruning a network's weights to a sparsity of 99 % for CIFAR-10 as well as SVHN. However, its performance drops drastically when increasing sparsity to 99.9 %. For SVHN, the models are even completely damaged, yielding a balanced accuracy of 10 %. Also for ImageNet, Robust-ADMM significantly harms the model robustness and natural performance. HARP, in turn, exceeds the performance of Robust-ADMM distinctively. Models pruned by our method preserve natural accuracy and robustness significantly better for high compression rates: At a sparsity of 99 %, the natural accuracy and the adversarial robustness is at a comparable level to the original model on CIFAR-10 and SVHN. Even for aggressive pruning to a sparsity of 99.9 %, the models show substantial resistance against adversarial inputs across training methods and considered attackers. On ImageNet, HARP outperforms Robust-ADMM in moderate and aggressive pruning with 20 percentage points for natural accuracy and 10 percentage points for robustness against various attacks.

Table 2: Comparing HARP with Robust-ADMM and HYDRA on models learned on CIFAR-10.

| | Adv. Training | Attack Type | Pre-Trained | 99 % Sparsity | | | | 99.9 % Sparsity | | | |
|---|---|---|---|---|---|---|---|---|---|---|---|
| | | | | R-ADMM | HYDRA | BCS-P | HARP | R-ADMM | HYDRA | BCS-P | HARP |
| ResNet18 | PGD | – | 82.89 | $71.42_{\pm0.37}$ | $75.53_{\pm1.95}$ | $71.84_{\pm0.35}$ | $\mathbf{80.13}_{\pm0.09}$ | $26.39_{\pm0.76}$ | $34.55_{\pm1.94}$ | $23.84_{\pm4.24}$ | $\mathbf{63.80}_{\pm0.47}$ |
| | | PGD | 50.05 | $42.31_{\pm0.31}$ | $45.84_{\pm1.20}$ | $45.63_{\pm0.45}$ | $\mathbf{50.41}_{\pm0.16}$ | $20.62_{\pm0.41}$ | $26.33_{\pm1.15}$ | $13.17_{\pm2.10}$ | $\mathbf{39.26}_{\pm0.19}$ |
| | | C&W$_\infty$ | 47.95 | $39.91_{\pm0.23}$ | $43.05_{\pm1.20}$ | $42.68_{\pm0.51}$ | $\mathbf{47.75}_{\pm0.17}$ | $20.00_{\pm0.62}$ | $26.08_{\pm1.39}$ | $12.57_{\pm2.26}$ | $\mathbf{36.19}_{\pm0.25}$ |
| | | APGD | 47.78 | $41.41_{\pm0.37}$ | $44.03_{\pm1.10}$ | $44.79_{\pm0.43}$ | $\mathbf{48.53}_{\pm0.24}$ | $20.26_{\pm0.39}$ | $25.57_{\pm1.23}$ | $12.63_{\pm2.32}$ | $\mathbf{38.05}_{\pm0.19}$ |
| | | AA | 45.30 | $37.64_{\pm0.33}$ | $40.89_{\pm0.60}$ | $40.55_{\pm0.42}$ | $\mathbf{45.58}_{\pm0.27}$ | $18.80_{\pm0.38}$ | $23.97_{\pm1.52}$ | $12.03_{\pm1.98}$ | $\mathbf{34.32}_{\pm0.30}$ |
| | TRADES | – | 81.30 | $70.73_{\pm0.56}$ | $75.07_{\pm0.45}$ | $71.77_{\pm0.63}$ | $\mathbf{77.90}_{\pm0.34}$ | $37.39_{\pm0.25}$ | $34.98_{\pm2.12}$ | $29.31_{\pm5.18}$ | $\mathbf{64.99}_{\pm0.47}$ |
| | | PGD | 53.21 | $41.34_{\pm0.33}$ | $46.60_{\pm0.26}$ | $44.69_{\pm0.58}$ | $\mathbf{50.25}_{\pm0.13}$ | $21.41_{\pm0.57}$ | $26.30_{\pm0.97}$ | $19.67_{\pm2.23}$ | $\mathbf{37.42}_{\pm0.30}$ |
| | | C&W$_\infty$ | 50.60 | $37.37_{\pm0.38}$ | $42.80_{\pm0.62}$ | $41.85_{\pm0.46}$ | $\mathbf{46.54}_{\pm0.07}$ | $20.53_{\pm0.50}$ | $27.19_{\pm1.28}$ | $20.07_{\pm2.13}$ | $\mathbf{33.11}_{\pm0.53}$ |
| | | APGD | 51.88 | $40.72_{\pm0.34}$ | $45.41_{\pm0.24}$ | $43.07_{\pm0.48}$ | $\mathbf{49.20}_{\pm0.13}$ | $21.19_{\pm0.85}$ | $25.63_{\pm1.38}$ | $19.58_{\pm2.18}$ | $\mathbf{36.61}_{\pm0.35}$ |
| | | AA | 49.48 | $36.69_{\pm0.37}$ | $41.99_{\pm0.63}$ | $41.54_{\pm0.37}$ | $\mathbf{45.53}_{\pm0.16}$ | $18.30_{\pm0.62}$ | $24.18_{\pm1.67}$ | $18.35_{\pm1.95}$ | $\mathbf{32.32}_{\pm0.56}$ |
| | MART | – | 80.16 | $74.14_{\pm0.31}$ | $72.88_{\pm1.36}$ | $66.18_{\pm0.89}$ | $\mathbf{75.51}_{\pm0.38}$ | $27.06_{\pm1.69}$ | $35.30_{\pm1.23}$ | $20.09_{\pm4.13}$ | $\mathbf{60.21}_{\pm0.19}$ |
| | | PGD | 53.72 | $42.12_{\pm0.42}$ | $49.39_{\pm1.31}$ | $47.23_{\pm0.28}$ | $\mathbf{51.21}_{\pm0.19}$ | $18.22_{\pm1.50}$ | $27.72_{\pm0.91}$ | $16.99_{\pm3.19}$ | $\mathbf{41.87}_{\pm0.24}$ |
| | | C&W$_\infty$ | 48.53 | $37.57_{\pm0.45}$ | $43.99_{\pm0.46}$ | $44.62_{\pm0.44}$ | $\mathbf{46.65}_{\pm0.16}$ | $19.89_{\pm1.61}$ | $26.45_{\pm1.76}$ | $17.04_{\pm3.19}$ | $\mathbf{36.36}_{\pm0.17}$ |
| | | APGD | 51.59 | $41.34_{\pm0.53}$ | $47.84_{\pm1.07}$ | $46.04_{\pm0.85}$ | $\mathbf{50.26}_{\pm0.18}$ | $18.04_{\pm1.47}$ | $26.91_{\pm1.04}$ | $16.67_{\pm3.28}$ | $\mathbf{40.82}_{\pm0.20}$ |
| | | AA | 46.39 | $36.68_{\pm0.48}$ | $42.08_{\pm0.58}$ | $43.23_{\pm0.51}$ | $\mathbf{44.89}_{\pm0.20}$ | $17.05_{\pm1.22}$ | $23.78_{\pm1.88}$ | $14.97_{\pm2.81}$ | $\mathbf{34.95}_{\pm0.26}$ |
| VGG16 | PGD | – | 79.68 | $62.28_{\pm0.42}$ | $67.33_{\pm1.30}$ | $70.07_{\pm0.42}$ | $\mathbf{78.21}_{\pm0.22}$ | $21.28_{\pm1.49}$ | $23.41_{\pm1.38}$ | $10.00_{\pm0.00}$ | $\mathbf{59.44}_{\pm0.28}$ |
| | | PGD | 47.60 | $37.54_{\pm0.51}$ | $41.47_{\pm0.96}$ | $44.70_{\pm0.39}$ | $\mathbf{48.59}_{\pm0.16}$ | $17.45_{\pm2.22}$ | $20.58_{\pm0.68}$ | $10.00_{\pm0.00}$ | $\mathbf{37.62}_{\pm0.31}$ |
| | | C&W$_\infty$ | 45.23 | $34.35_{\pm0.59}$ | $38.09_{\pm0.88}$ | $41.35_{\pm0.52}$ | $\mathbf{45.15}_{\pm0.33}$ | $17.46_{\pm2.32}$ | $20.99_{\pm0.92}$ | $10.00_{\pm0.00}$ | $\mathbf{33.93}_{\pm0.39}$ |
| | | APGD | 45.05 | $36.38_{\pm0.74}$ | $39.84_{\pm0.96}$ | $42.48_{\pm0.37}$ | $\mathbf{46.61}_{\pm0.19}$ | $16.84_{\pm2.26}$ | $20.32_{\pm0.49}$ | $10.00_{\pm0.00}$ | $\mathbf{36.11}_{\pm0.35}$ |
| | | AA | 42.12 | $32.21_{\pm0.82}$ | $36.15_{\pm0.53}$ | $38.61_{\pm0.40}$ | $\mathbf{42.48}_{\pm0.37}$ | $16.29_{\pm2.48}$ | $19.74_{\pm0.40}$ | $10.00_{\pm0.00}$ | $\mathbf{32.16}_{\pm0.39}$ |
| | TRADES | – | 80.18 | $64.59_{\pm0.32}$ | $68.34_{\pm1.33}$ | $71.45_{\pm0.71}$ | $\mathbf{76.91}_{\pm0.31}$ | $22.76_{\pm1.04}$ | $28.23_{\pm0.43}$ | $10.00_{\pm0.00}$ | $\mathbf{62.46}_{\pm0.69}$ |
| | | PGD | 49.72 | $35.69_{\pm0.58}$ | $40.61_{\pm0.68}$ | $43.18_{\pm0.65}$ | $\mathbf{47.79}_{\pm0.18}$ | $15.60_{\pm0.42}$ | $20.52_{\pm0.68}$ | $10.00_{\pm0.00}$ | $\mathbf{34.32}_{\pm0.49}$ |
| | | C&W$_\infty$ | 46.59 | $31.90_{\pm0.46}$ | $36.66_{\pm1.06}$ | $41.11_{\pm0.99}$ | $\mathbf{43.85}_{\pm0.18}$ | $15.91_{\pm0.56}$ | $20.71_{\pm0.92}$ | $10.00_{\pm0.00}$ | $\mathbf{29.85}_{\pm0.60}$ |
| | | APGD | 47.89 | $34.91_{\pm0.59}$ | $39.39_{\pm0.43}$ | $43.89_{\pm0.89}$ | $\mathbf{46.32}_{\pm0.28}$ | $14.63_{\pm0.57}$ | $20.04_{\pm0.62}$ | $10.00_{\pm0.00}$ | $\mathbf{33.26}_{\pm0.50}$ |
| | | AA | 45.09 | $30.99_{\pm0.51}$ | $35.67_{\pm0.88}$ | $39.88_{\pm0.61}$ | $\mathbf{42.65}_{\pm0.16}$ | $13.17_{\pm0.66}$ | $18.46_{\pm0.71}$ | $10.00_{\pm0.00}$ | $\mathbf{29.21}_{\pm0.50}$ |
| | MART | – | 73.44 | $66.24_{\pm0.53}$ | $64.69_{\pm0.74}$ | $67.85_{\pm0.55}$ | $\mathbf{73.49}_{\pm0.42}$ | $25.01_{\pm1.56}$ | $19.33_{\pm1.71}$ | $10.00_{\pm0.00}$ | $\mathbf{56.85}_{\pm0.94}$ |
| | | PGD | 51.51 | $35.20_{\pm0.90}$ | $44.18_{\pm1.53}$ | $47.65_{\pm0.60}$ | $\mathbf{51.03}_{\pm0.13}$ | $17.95_{\pm2.34}$ | $17.95_{\pm0.89}$ | $10.00_{\pm0.00}$ | $\mathbf{40.21}_{\pm0.31}$ |
| | | C&W$_\infty$ | 44.38 | $30.75_{\pm0.47}$ | $38.20_{\pm0.34}$ | $41.68_{\pm0.41}$ | $\mathbf{44.34}_{\pm0.14}$ | $18.49_{\pm2.20}$ | $17.60_{\pm0.78}$ | $10.00_{\pm0.00}$ | $\mathbf{33.34}_{\pm0.45}$ |
| | | APGD | 49.56 | $33.83_{\pm0.96}$ | $42.47_{\pm1.21}$ | $46.53_{\pm0.71}$ | $\mathbf{48.94}_{\pm0.23}$ | $16.93_{\pm2.77}$ | $17.23_{\pm0.80}$ | $10.00_{\pm0.00}$ | $\mathbf{38.56}_{\pm0.32}$ |
| | | AA | 42.20 | $29.48_{\pm0.74}$ | $36.45_{\pm0.59}$ | $40.73_{\pm0.61}$ | $\mathbf{42.20}_{\pm0.15}$ | $15.54_{\pm2.81}$ | $16.14_{\pm0.92}$ | $10.00_{\pm0.00}$ | $\mathbf{31.74}_{\pm0.41}$ |

Table 3: Comparing HARP with Robust-ADMM and HYDRA on models learned on SVHN.

| Adv. Training | Attack Type | Pre-Trained | 99 % Sparsity | | | | 99.9 % Sparsity | | | |
|---|---|---|---|---|---|---|---|---|---|---|
| | | | R-ADMM | HYDRA | BCS-P | HARP | R-ADMM | HYDRA | BCS-P | HARP |
| **ResNet18** PGD | – | 91.92 | 84.34±1.44 | 88.71±0.49 | 88.39±0.79 | **90.70**±0.79 | 10.00±0.00 | 36.00±1.34 | 10.00±0.00 | **81.55**±0.45 |
| | PGD | 56.82 | 43.55±1.73 | 50.56±0.51 | 50.61±0.50 | **57.21**±1.96 | 10.00±0.00 | 19.69±0.32 | 10.00±0.00 | **42.38**±0.24 |
| | C&W$_\infty$ | 52.41 | 38.38±0.96 | 45.53±0.64 | 45.01±0.46 | **51.37**±1.44 | 10.00±0.00 | 20.17±1.36 | 10.00±0.00 | **37.72**±0.14 |
| | APGD | 47.67 | 40.15±0.87 | 46.92±0.28 | 46.70±0.38 | **47.40**±1.27 | 10.00±0.00 | 18.48±0.32 | 10.00±0.00 | **38.80**±0.16 |
| | AA | 43.56 | 35.47±0.77 | 42.40±0.28 | **42.60**±0.26 | 42.07±2.10 | 10.00±0.00 | 16.49±0.60 | 10.00±0.00 | **34.31**±0.32 |
| TRADES | – | 90.70 | 79.35±1.45 | 86.07±1.97 | 86.24±1.10 | **88.61**±1.38 | 11.68±3.77 | 29.81±3.70 | 20.25±9.68 | **73.33**±0.99 |
| | PGD | 57.12 | 40.92±1.55 | 49.10±0.81 | 50.66±0.84 | **57.80**±1.56 | 10.55±1.24 | 15.17±3.20 | 15.60±6.39 | **37.43**±0.20 |
| | C&W$_\infty$ | 51.40 | 35.31±1.67 | 43.93±0.83 | 45.32±0.71 | **50.86**±1.47 | 10.42±0.95 | 14.07±1.99 | 14.35±5.52 | **32.74**±0.13 |
| | APGD | 45.60 | 37.81±1.88 | 45.49±0.23 | **45.92**±0.63 | 43.64±1.64 | 10.49±1.11 | 14.37±3.74 | 15.57±6.36 | **34.85**±0.24 |
| | AA | 42.08 | 32.95±1.49 | 40.68±0.39 | **43.20**±0.83 | 36.42±2.42 | 10.35±0.79 | 12.52±2.81 | 13.27±4.57 | **30.71**±0.29 |
| MART | – | 90.84 | 85.39±0.82 | 86.16±1.65 | 87.17±0.72 | **88.29**±0.95 | 25.18±8.65 | 35.32±4.04 | 10.00±0.00 | **74.99**±1.82 |
| | PGD | 59.53 | 44.13±1.21 | 50.87±0.82 | 51.21±0.76 | **57.25**±2.52 | 14.40±2.80 | 19.93±1.92 | 10.00±0.00 | **41.16**±0.76 |
| | C&W$_\infty$ | 51.48 | 36.50±1.28 | 43.45±0.78 | 45.49±0.59 | **48.77**±1.50 | 13.30±2.28 | 20.95±1.61 | 10.00±0.00 | **34.46**±0.61 |
| | APGD | 51.25 | 40.13±1.16 | 47.88±0.64 | 47.24±0.49 | **50.55**±1.69 | 13.51±2.21 | 18.66±1.79 | 10.00±0.00 | **38.27**±0.69 |
| | AA | 42.93 | 34.29±1.34 | 40.08±0.42 | 41.26±0.53 | **41.90**±2.88 | 11.50±1.13 | 15.79±1.37 | 10.00±0.00 | **31.92**±0.30 |
| **VGG16** PGD | – | 91.38 | 76.33±2.16 | 85.62±0.46 | 84.72±0.55 | **88.97**±0.71 | 10.00±0.00 | 21.79±3.10 | 10.00±0.00 | **80.29**±0.68 |
| | PGD | 52.22 | 37.08±0.93 | 45.31±0.55 | 46.55±0.52 | **50.45**±0.45 | 10.00±0.00 | 13.75±0.35 | 10.00±0.00 | **40.85**±0.60 |
| | C&W$_\infty$ | 47.74 | 30.80±0.89 | 39.42±0.58 | 41.70±0.43 | **45.54**±0.48 | 10.00±0.00 | 13.95±0.51 | 10.00±0.00 | **35.66**±0.15 |
| | APGD | 47.72 | 33.22±0.93 | 41.19±0.49 | 43.71±0.54 | **46.59**±0.72 | 10.00±0.00 | 12.57±0.83 | 10.00±0.00 | **37.17**±0.44 |
| | AA | 43.59 | 27.39±1.16 | 36.24±0.48 | 39.62±0.70 | **42.36**±0.88 | 10.00±0.00 | 11.99±0.59 | 10.00±0.00 | **32.47**±0.57 |
| TRADES | – | 88.66 | 74.10±2.12 | 81.87±2.61 | 84.13±0.74 | **84.69**±0.94 | 10.00±0.00 | 14.67±5.58 | 10.00±0.00 | **75.78**±0.88 |
| | PGD | 52.90 | 35.37±0.79 | 43.92±0.94 | 47.79±0.83 | **50.11**±0.47 | 10.00±0.00 | 10.94±1.33 | 10.00±0.00 | **37.97**±0.40 |
| | C&W$_\infty$ | 47.06 | 29.63±0.79 | 37.94±0.71 | 42.91±0.86 | **43.85**±0.45 | 10.00±0.00 | 11.37±1.86 | 10.00±0.00 | **32.56**±0.18 |
| | APGD | 49.87 | 31.45±0.61 | 40.32±0.89 | 44.83±0.86 | **46.33**±0.55 | 10.00±0.00 | 10.23±0.58 | 10.00±0.00 | **34.88**±0.40 |
| | AA | 44.62 | 27.00±0.46 | 35.46±1.01 | 40.64±0.72 | **41.32**±0.62 | 10.00±0.00 | 10.07±0.44 | 10.00±0.00 | **30.59**±0.40 |
| MART | – | 87.45 | 80.25±1.37 | 81.51±2.67 | 82.98±1.04 | **84.09**±0.51 | 10.00±0.00 | 17.02±3.95 | 10.00±0.00 | **70.53**±1.07 |
| | PGD | 52.20 | 39.07±2.11 | 46.00±0.87 | 46.64±0.74 | **49.93**±0.84 | 10.00±0.00 | 13.28±2.22 | 10.00±0.00 | **37.56**±0.79 |
| | C&W$_\infty$ | 44.98 | 31.07±2.21 | 37.53±0.87 | 39.90±0.81 | **42.70**±0.74 | 10.00±0.00 | 13.10±2.18 | 10.00±0.00 | **31.02**±0.75 |
| | APGD | 46.30 | 33.24±1.64 | 38.88±0.85 | 41.23±0.77 | **43.73**±0.68 | 10.00±0.00 | 11.29±1.82 | 10.00±0.00 | **33.42**±1.28 |
| | AA | 40.43 | 26.69±1.09 | 31.31±0.95 | 33.00±0.97 | **37.39**±1.07 | 10.00±0.00 | 11.13±1.64 | 10.00±0.00 | **27.18**±1.05 |

Table 4: Comparing HARP with Robust-ADMM and HYDRA on ResNet50 models for ImageNet.

| Attack | FREE-AT | 90 % Sparsity | | | 99 % Sparsity | | |
|---|---|---|---|---|---|---|---|
| | | R-ADMM | HYDRA | HARP | R-ADMM | HYDRA | HARP |
| – | 60.25 | 35.26±0.46 | 49.44±0.37 | **55.21**±0.36 | 11.41±0.32 | 27.00±0.66 | **34.62**±0.36 |
| PGD | 32.82 | 14.35±0.41 | 23.75±0.33 | **27.10**±0.41 | 5.15±0.17 | 12.23±0.19 | **14.67**±0.32 |
| C&W$_\infty$ | 30.67 | 12.35±0.33 | 21.60±0.27 | **24.62**±0.38 | 4.03±0.22 | 11.22±0.18 | **12.42**±0.33 |
| APGD | 31.54 | 13.53±0.39 | 23.14±0.27 | **25.57**±0.33 | 4.85±0.31 | 12.34±0.34 | **13.47**±0.34 |
| AA | 28.79 | 11.01±0.25 | 19.88±0.29 | **22.57**±0.41 | 3.69±0.35 | 10.09±0.40 | **11.24**±0.43 |

**Comparison to HYDRA.** Sehwag et al. (2020) demonstrate remarkable performance on robustly pruning a model's weights, which we are able to confirm in our experiments. The results are presented in Tables 2 to 4 for CIFAR-10, SVHN, and ImageNet. HYDRA consistently improves upon Robust-ADMM for the sparsity levels considered in our experiments. However, while HYDRA maintains its natural accuracy and robustness for a sparsity of 99 % well, for 99.9 % its performance degrades heavily (similarly to Robust-ADMM). Differently, HARP is able to maintain both accuracy metrics significantly better for aggressive pruning. In most cases, our method doubles the accuracy of HYDRA at a sparsity of 99.9 %, yielding an unprecedentedly competitive performance for such highly compressed networks. Also, the additionally gained robustness stemming from using MART-AT rather than TRADES-AT nicely transfers to models pruned by HARP, while HYDRA cannot benefit from this advantage at the same level. For ImageNet, HARP is also able to outperform HYDRA noticeably and has the highest effectiveness in suppressing the model performance recession. However, the drop in performance from 90 % to 99 % sparsity is comparable to HYDRA and remains significant. Limiting the decay even further for large-scale datasets remains a challenge.

**Comparison to BCS-P.** Similar to our approach, BCS-P (Özdenizci & Legenstein, 2021) strives for a non-uniform compression strategy. To do so, they introduce "Bayesian Connectivity Sampling" for selecting important network elements from a global view. In contrast to HARP, it however does not follow the staged pruning-pipeline of Han et al. (2015) but achieves robust pruning via an end-to-end sparse training-procedure. In Fig. 1, we present results of BCS-P for a VGG16 model trained with PGD-AT on CIFAR-10, and explicitly investigate its performance at sparsity 99 % and 99.9 % on CIFAR-10 and SVHN in Tables 2 and 3. While it does not allow for channel pruning, BCS-P outperforms both HYDRA and Robust-ADMM in pruning VGG16's weights, while remains inferior to HYDRA in pruning ResNet18. HARP, however, surpasses BCS-P significantly with

increasing model sparsity and yields a functional model at 99.9 % sparsity, while BCS-P does not. In Section 4.3, we analyze the strategies and attribute the observed collapse to a heavily pruned input layer. More comparative results and experiments on ResNet18 are presented in Appendix A.2.

**Comparison to MAD.** Lee et al. (2022) propose to compute an adversarial saliency map of a robustly pre-trained model via "Kronecker-Factored Approximate Curvature" (Martens & Grosse, 2015) and mask out weights that are least relevant for robustness. Unfortunately, we have not been able to fully reproduce the MAD's results. We thus present the numbers reported by the authors of MAD along-side results for HARP in the experimental setup described in the original publication and adjust pre-training accordingly for CIFAR-10 and SVHN. Lee et al. (2022) use a PGD step-size of 0.0069 for adversarial pre-training over 60 epochs and a $2 \times 10^{-4}$ weight decay. At moderate compression ($< 90\,\%$ sparsity) shown in Table 5, MAD and HARP are on a par. Neither one harms the performance of the neural network for CIFAR-10 and SVHN.

As indicated in Fig. 3, showing the PGD-10 adversarial robustness on CIFAR-10 and SVHN, this changes for 99 % sparsity. While our method maintains robustness, MAD drops in performance significantly. In particular, it is noteworthy that at a sparsity of 99.9 % HARP achieves similar adversarial robustness than MAD at a sparsity of 99 %, meaning, the network pruned by MAD is an order of magnitude larger.

Table 5: Pruning 90 % weights on pre-trained robust neural networks via HARP and MAD.

| Model | Attack | CIFAR-10 | | | SVHN | | |
|---|---|---|---|---|---|---|---|
| | | Pre-train | MAD | HARP | Pre-train | MAD | HARP |
| VGG16 | – | 81.32 | 81.44 | **82.10** | 93.19 | 92.41 | **92.38** |
| | PGD-10 | 50.21 | **51.75** | 50.94 | 56.03 | **60.42** | 58.97 |
| ResNet18 | – | 84.22 | 82.73 | **84.76** | 93.67 | 92.63 | **92.80** |
| | PGD-10 | 53.33 | 52.98 | **53.46** | 59.62 | **60.61** | 60.37 |

Figure 3: Comparing PGD-10 adversarial robustness of Robust-ADMM, HYDRA, MAD, and HARP on VGG16 models learned on CIFAR-10. All pre-trained models are provided by Lee et al. (2022).

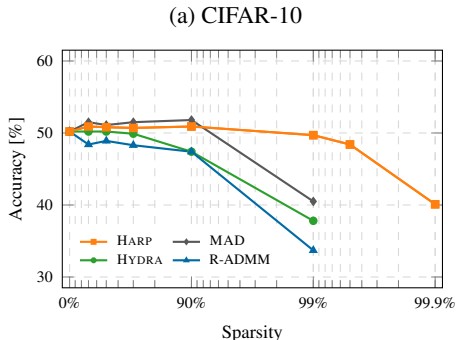
(a) CIFAR-10

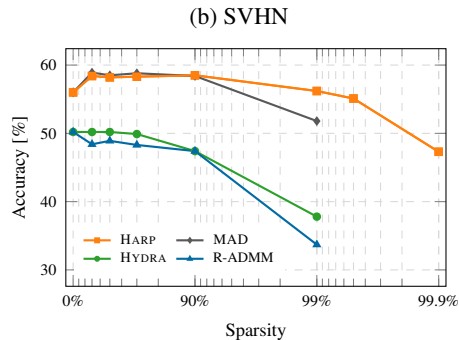
(b) SVHN

## 4.3 STRATEGY ANALYSIS

Also in conventional network pruning, non-uniform compression strategies have been proven effective, for instance, ERK by Evci et al. (2020) and LAMP by Lee et al. (2021). While we are the first to show an equivalent benefit for adversarial robust pruning, in this section, we investigate (a) if these approaches can also be applied to adversarial robust pruning, and (b) if they find similar compression strategies in comparison to HARP. Both ERK and LAMP focus on model parameters and support weight pruning only. We adjust HYDRA to accept different compression rates per layer and

Table 6: Comparing performance improvement of Robust-ADMM and HYDRA by using ERK and LAMP and by HARP on CIFAR-10. Natural accuracy and PGD-10 robustness are presented left and right of the / character.

| Model | Sparsity | R-ADMM | | | HYDRA | | | HARP |
|---|---|---|---|---|---|---|---|---|
| | | Original | w/ ERK | w/ LAMP | Original | w/ ERK | w/ LAMP | |
| ResNet18 | 99 % | 71.42 / 42.31 | 80.36 / **48.38** | **80.64** / 48.28 | 75.53 / 45.84 | 79.09 / 49.17 | **80.16** / **50.07** | **80.25** / 50.36 |
| | 99.9 % | 26.39 / 20.62 | 54.51 / 33.06 | **57.16** / **34.05** | 34.55 / 26.08 | 55.73 / 35.09 | **57.07** / **35.91** | 63.99 / 39.39 |
| VGG16 | 99 % | 62.28 / 37.54 | 70.33 / 43.30 | **74.38** / **46.39** | 67.33 / 41.47 | 72.19 / 45.05 | **76.75** / **47.96** | 78.58 / 48.71 |
| | 99.9 % | 21.28 / 17.46 | 43.35 / 29.11 | **48.96** / **32.39** | 23.41 / 20.99 | 50.38 / 34.32 | **57.93** / **36.01** | 59.13 / 37.36 |

assign the strategies determined by ERK and LAMP. The results for pruning a PGD-AT pre-trained VGG16 model are presented in Table 6. LAMP in particular produces very promising compression strategies that, however, are surpassed by HARP. For ImageNet learned with ResNet50, in turn, using LAMP and ERK with HYDRA impacts performance negatively, unlike HARP (cf. Table 11 in Appendix A.3). In Fig. 4, we present the non-uniform compression strategies determined by the different approaches as (a) compression rates and (b) overall preserved parameters per layer. Additionally, we show a uniform compression yielding 99.9 % sparsity as reference. Note, that due to the layer's difference in size the absolute number of preserved parameters varies for the uniform strategy in Fig. 4b also. It is interesting to see that LAMP puts more focus on layers in the middle sacrificing input/output information, while HARP tends to focus on front and back layer stronger. ERK, in turn, even maintains large portions of fc1. Interestingly, BCS-P yields an almost uniform strategy although it strives for non-uniformity. Even worse, only two out of 1,728 parameters in VGG16's input layer are preserved at 99.9 % sparsity, severely impacting performance.

Figure 4: Strategy comparison for pruning a VGG16 with target 99.9 % sparsity on CIFAR-10.

(a) Layer-wise compression rates

(b) Layer-wise preserved parameters

## 5 CONCLUSIONS

For pruning neural networks it is crucial to decide *how many and which parameters to prune*. This does not only affect natural accuracy, but also the robustness against adversarial input manipulations. Maintaining adversarial robustness is crucial for safety-critical applications, such as autonomous driving and edge AI. Our method, HARP, incorporates a global view of the network's compression and scores for gauging the importance of network connections into a dynamically regularized loss formulation that allows to significantly outperform related work. We are the first to reach competitive performance for highly aggressive pruning, aiming at up to 99.9 % network sparsity. Therewith, we show that learning a global, but layer-specific and, thus, non-uniform compression strategy is at least as important as deciding on what connections to prune.

**Limitations.** Similarly to the majority of related work in the field, our method is of empirical nature and, thus, can benefit from future work on theoretic analyses of how and why our pruning approach arrives at a particular compression strategy. In our evaluation, we show that HARP's compression is close to the theoretically founded strategies of LAMP, raising the hope that a rigorous theoretic justification is possible. Ideally this extends to a theoretically determined, one-shot calculation that spares the computational effort of the optimization problem considered for HARP.

## ACKNOWLEDGEMENTS

The authors thank the anonymous reviewers for their valuable suggestions, and gratefully acknowledge funding by the Helmholtz Association (HGF) within topic "46.23 Engineering Secure Systems" and by SAP S.E. under project DE-2020-021.

## ETHICS STATEMENT

Deep neural networks are used in a wide variety of applications. While these applications may have ethical implications, our method itself does not. HARP aims to compress an existing model, maintaining accuracy and resistance against input manipulation attacks. As such, our method is agnostic to the underlying application. Moreover, we do not reveal any vulnerabilities in the robust model compression, do not use any personalized data, and do not involve human subjects in our experiments.

## REPRODUCIBILITY STATEMENT

For the sake of reproducibility and to foster future research, we make the implementations of HARP for holistic adversarially robust pruning publicly available at:

https://intellisec.de/research/harp

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

Table 7: Overview of used method abbreviations.

| Category | Used Abbreviation | Reference |
|---|---|---|
| Attacks | PGD | Madry et al. (2018) |
| | C&W$_\infty$ | Carlini & Wagner (2017) |
| | APGD, AUTOPGD | Croce & Hein (2020a) |
| | AA, AUTOATTACK | Croce & Hein (2020a) |
| Adversarial Training | PGD-AT | Madry et al. (2018) |
| | TRADES-AT | Zhang et al. (2019) |
| | MART-AT | Wang et al. (2020) |
| Non-uniform compression | ERK | Evci et al. (2020) |
| | LAMP | Lee et al. (2021) |
| Adversarially robust pruning | HYDRA | Sehwag et al. (2020) |
| | Robust-ADMM, R-ADMM | Ye et al. (2019) |
| | BCS-P | Özdenizci & Legenstein (2021) |
| | MAD | Lee et al. (2022) |
| | DNR | Kundu et al. (2021) |
| | HARP (Ours) | Zhao & Wressnegger (2023) |

# A APPENDIX

We proceed to analyze the influence of different step sizes to regularize HARP (Appendix A.1), before we extend upon the empirical evidence of our methods performance: We broaden our comparison to related work by inspecting the performance of BCS-P and MAD on ResNet18 and additionally considering DNR (Appendix A.2). We also extend upon the analysis of HARP's pruning strategies (Appendix A.3), inspect the parameter distribution of the pruned models (Appendix A.4, and evaluate HARP's performance based on natural training (Appendix A.5). Moreover, we demonstrate an extension of our method to channel/structural pruning (Appendix A.6) and present a comparison of training consumption for all methods in our evaluation (Appendix A.7).

## A.1 INFLUENCE OF DYNAMIC $\gamma$ ON HARP'S PRUNING

In this section, we present results for different regularization step-sizes of HARP when pruning VGG16 to a sparsity of 99.9 %. As illustrated in Figs. 5a and 5b, HARP exhibits strong oscillation for small step sizes ($\gamma \leq 0.1$), while larger steps ($\gamma = 1.0$) restrict performance improvements. A step size of $\gamma = 0.01$, in turn, helps the model to better converge. HARP performs similarly for a slightly changed step size of $\gamma = 0.005$, but yielding the target compression is delayed. When

Figure 5: Different $\gamma$ step-size for pruning 99.9 % weights of a VGG16 by HARP on CIFAR-10.

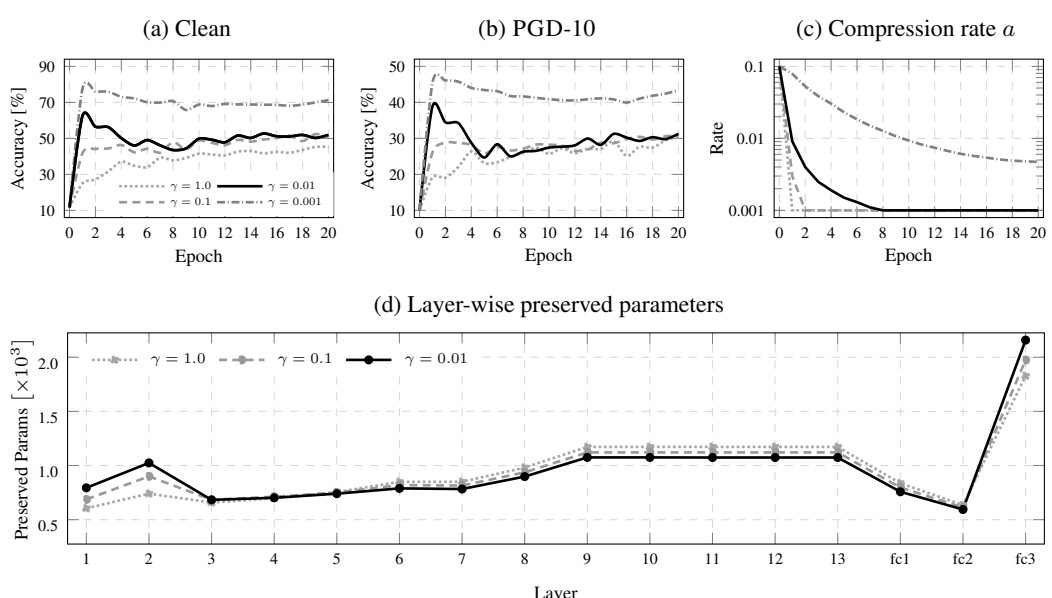

(a) Clean  (b) PGD-10  (c) Compression rate $a$

(d) Layer-wise preserved parameters

lowering the step size further to $0.001$, the target compression cannot be reached anymore as shown in Fig. 5c. Hence, small step sizes shorten the time for adversarial training but cannot guarantee yielding the target compression. In Fig. 5d, we summarize the generated strategies that satisfy the target compression. With decreasing $\gamma$, HARP preserves layer in front and at the end stronger. Model parameters in the middle layers are sacrificed to preserve higher performance.

## A.2 EXTENDED COMPARISON TO RELATED WORK

**Comparison to BCS-P.** Fig. 6 shows extended experiments for ResNet18. At sparsity $99.9\%$, model collapse appears after BCS-P's pruning, particularly on SVHN. Fig. 7 visualizes pruning strategies for ResNet18 on SVHN. Similar to Fig. 4a, BCS-P exhibits an almost uniform pruning strategy, compressing the input layer down to $0.0005$. This reduction refers to one out of $1{,}728$ parameters which clearly cannot yield any meaningful prediction. Additionally, we extend to initialize BCS-P with LAMP non-uniform strategy (cf. Table 8). Similar to the effect on HYDRA, BCS-P significantly benefit from the non-uniform initialization. Especially, the plight of BCS-P at sparsity $99.9\%$ is solved. However, HYDRA shows a better adaptability to LAMP's initialization. At the same time, HARP remains overall the best performance on both $99\%$ and $99.9\%$ sparsity.

Figure 6: Comparing HARP with BCS-P for pruning ResNet18 weights on CIFAR-10 and SVHN.

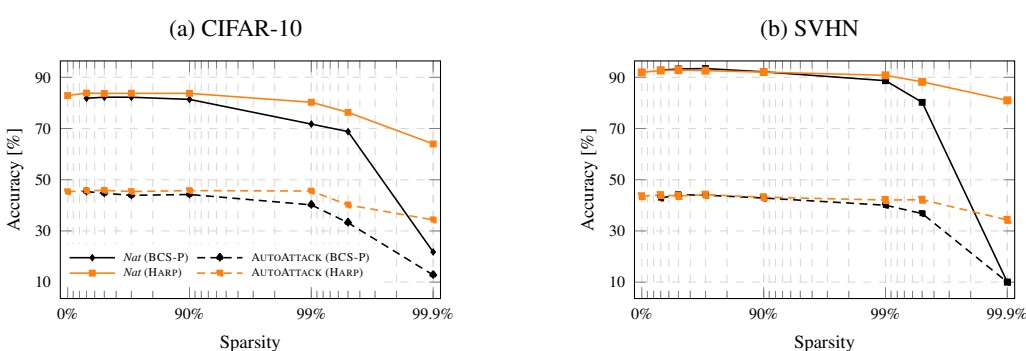

Figure 7: Strategy comparison for pruning a ResNet18 with target $99.9\%$ sparsity on SVHN.

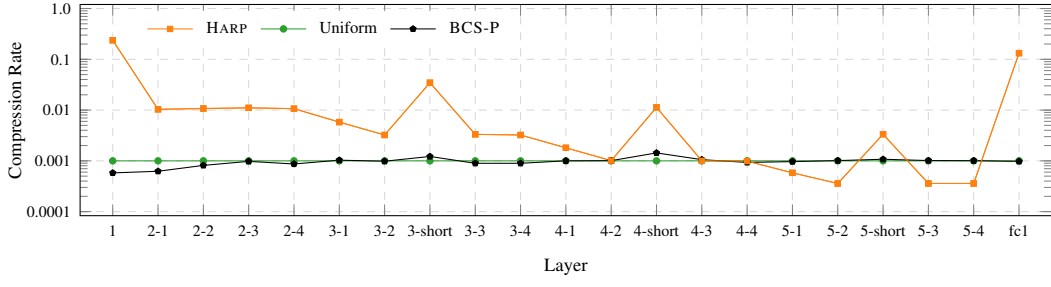

Table 8: LAMP initialization on HYDRA and BCS-P and their comparison to HARP on CIFAR-10. The natural accuracy and PGD-10 robustness are presented left and right of the / character.

| Model | Sparsity | HYDRA | | BCS-P | | HARP |
|---|---|---|---|---|---|---|
| | | Original | w/ LAMP | Original | w/ LAMP | |
| ResNet18 | 99 % | 75.53 / 45.84 | **80.16 / 50.07** | 71.73 / 45.32 | **72.58 / 45.42** | **80.25 / 50.36** |
| | 99.9 % | 34.55 / 26.08 | **57.07 / 35.91** | 21.78 / 12.78 | **56.33 / 35.21** | **63.99 / 39.39** |
| VGG16 | 99 % | 67.33 / 41.47 | **76.75 / 47.96** | 69.81 / 42.26 | **70.47 / 43.71** | **78.58 / 48.71** |
| | 99.9 % | 23.41 / 20.99 | **57.93 / 36.01** | 10.00 / 10.00 | **43.77 / 30.34** | **59.13 / 37.36** |

**Comparison to MAD.** We maintain the same training setting in Fig. 3. Similarly, MAD presents a promising performance on SVHN (cf Fig. 8) to further enhance adversarial resistance by mod-

erately pruning. At the same time, HARP presents robustness improvement as well in moderate pruning. In distinction from MAD, however, HARP remains the stable performance on pruning a higher sparsity while MAD performs less efficiently at aggressive pruning.

Figure 8: Comparing PGD-10 adversarial robustness of ResNet18 learned on SVHN and pruned by Robust-ADMM, HYDRA, MAD and HARP. All pre-trained robust models are from MAD.

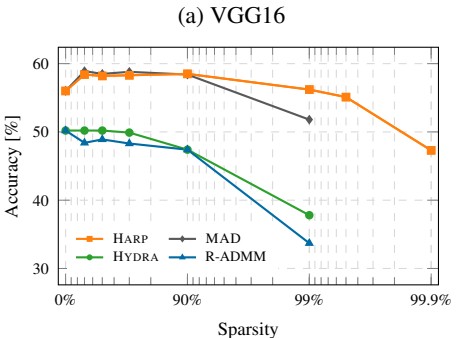

(a) VGG16

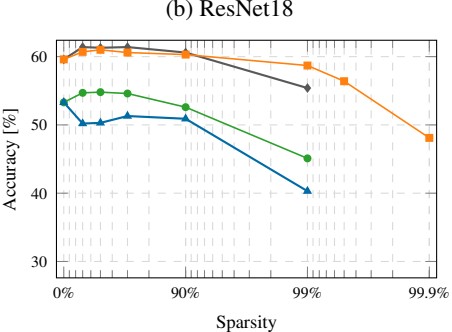

(b) ResNet18

**Comparison to DNR.** Kundu et al. (2021) propose a robust pruning framework, DNR, to learn compact and robust neural network through dynamic network rewiring. To conduct the comparison, we follow the authors training setting (Rakin et al., 2018) and use the authors evaluation metrics. Results are reported in Table 9. At a high compression (50×) HARP plays its strengths. For CIFAR-10, our method yields robustness scores comparable to DNR at a compression of $\approx 20\times$. The natural accuracy is slightly inferior, though. For the large-scale dataset, Tiny-ImageNet (Le & Yang, 2015), in turn, HARP yields higher robustness than DNR for compression up to $50\times$, but falls behind in natural accuracy again. In summary, HARP yields higher robustness while DNR seems favors natural accuracy.

Table 9: Comparing HARP with DNR in pruning neural network weights.

| Dataset | Model | Method | Compression | Sparsity [%] | Natural | FGSM | PGD-7 |
|---|---|---|---|---|---|---|---|
| CIFAR-10 | VGG16 | DNR | 20.85 × | 95.4 | **86.74** | 52.92 | 43.21 |
| | | HARP | 20.85 × | 95.4 | 85.09 | **53.55** | **45.34** |
| | | | 50.00 × | 98.0 | 83.16 | 52.27 | 45.13 |
| | | | 100.00 × | 99.0 | 81.35 | 50.83 | 42.75 |
| | ResNet18 | DNR | 21.57 × | 95.2 | **87.32** | 55.13 | 47.35 |
| | | HARP | 21.57 × | 95.2 | 86.97 | **56.33** | **49.31** |
| | | | 50.00 × | 98.0 | 84.30 | 53.87 | 46.78 |
| | | | 100.00 × | 99.0 | 82.35 | 52.06 | 45.47 |
| Tiny-ImageNet | VGG16 | DNR | 20.63 × | 95.2 | **51.71** | 18.21 | 14.46 |
| | | HARP | 20.63 × | 95.2 | 50.35 | **19.06** | **15.84** |
| | | | 50.00 × | 98.0 | 50.21 | **19.61** | **16.24** |

### A.3 EXTENDED STRATEGY ANALYSIS

In this section, we investigate the impact of non-uniform strategies on robust weight pruning. Firstly, we compare different strategies used to initialize HARP's pruning in order to present the insensibility on HARP' layer-wise rate initialization. In the next step, we analyze the end-to-end learning process with regard to the stability of training with and without using non-uniform strategies on HYDRA. Secondly, we assess HYDRA's performance by using ERK and LAMP for ImageNet, demonstrating that the positive impact of the strategies' non-uniformity is not consistent for different datasets.

**Analysis of HARP's strategy initialization.** As demonstrated by Sehwag et al. (2020), score initialization is key to realize a improvement on the weight pruning. In Fig. 9, we provide the learning processes of HARP's pruning at sparsity that are initialized by ERK, LAMP and the default uniform strategy. In comparison with ERK and LAMP, uniform strategy leads to a dramatic performance degradation after initialization. However, the training on scores $S_W$ and rates $r$ shows the cape of

Figure 9: Processes of weight pruning on a robust ResNet18 for CIFAR-10 by HARP with different initialization strategies: default Uniform (orange), ERK (gray) and LAMP (blue). Natural performance and PGD-10 adversarial robustness are presented as solid lines and dashed lines, respectively.

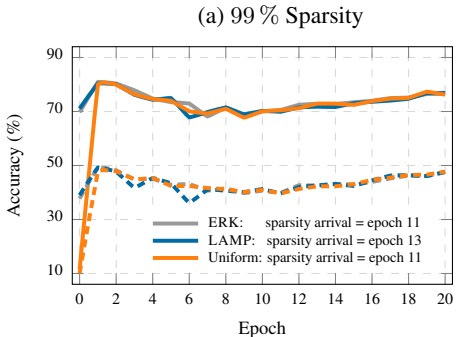

(a) 99 % Sparsity

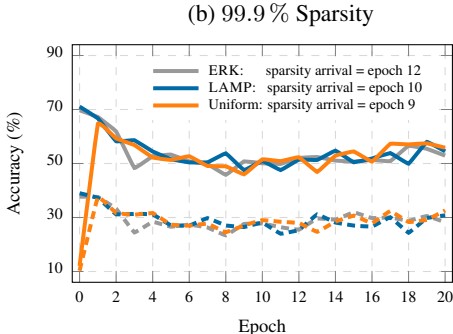

(b) 99.9 % Sparsity

recovering the network performance, resulting in a largely similar network performance to ERK and LAMP after the first epoch training. In the subsequent epochs, the learning curves on different initializations are essentially overlap. By further evaluating the final pruned networks after fine-tuning, results in Table 10 show that the difference between uniform adn non-uniform strategies are negligible for the layer compression rate initialization. Confirming, that HARP is insensitive to different strategies initialization. Therewith, we rely on the uniform strategy as the default setting in our experiments.

Table 10: Comparing different strategies initialization on HARP for CIFAR-10.

| Model | Sparsity | Uniform | ERK | LAMP |
|---|---|---|---|---|
| ResNet18 | 99 % | 80.25 / 50.36 | 79.42 (−0.83) / 50.43 (+0.07) | 80.48 (+0.23) / 50.33 (−0.03) |
| | 99.9 % | 63.99 / 39.39 | 63.61 (−0.38) / 39.20 (−0.19) | 63.64 (−0.35) / 39.43 (+0.04) |
| VGG16 | 99 % | 78.50 / 48.71 | 78.38 (−0.12) / 48.40 (+0.31) | 78.82 (+0.32) / 48.69 (−0.02) |
| | 99.9 % | 59.33 / 37.46 | 59.79 (+0.46) / 37.72 (+0.26) | 59.85 (+0.52) / 37.65 (+0.19) |

**Comparison of end-to-end compression pipelines.** In Fig. 10, we presents a comparison between different strategies on HYDRA and HARP across the end-to-end compression pipeline. For pruning 99 % weights, the potential of the original HYDRA is impaired by the use of a uniform strategy in the pruning phase, limiting the potential for improving via fine-tuning. At the same time, the non-uniform strategies ERK and LAMP prove their positive impact in the pruning stage. Without fine-tuning, they achieve already better results than HYDRA. In HARP's pruning, we observe a down-up performance curve, indicating that the network performance recovers after the robust networks is compressed. Further improved by fine-tuning, HARP achieves slightly better results than HYDRA with ERK and LAMP. In pruning a 99.9 % sparsity, HYDRA produces a dramatic decrease in performance that can, however, be compensated by using LAMP and ERK. Using HARP during pruning results in the highest network performance, approximating the final result after fine-tuning.

**Comparing non-uniform strategies for ImageNet.** Presented in Table 6, HYDRA is significantly improved by applying ERK and LAMP. In Table 11, we further investigate their impact on the large-scale dataset ImageNet. Compared to original HYDRA, using ERK and LAMP strategies leads to a performance degradation, and with LAMP yields more harm than ERK at sparsity 99 %. Fig. 11 visualizes the strategies of sparsity 99 % for ImageNet experiments. Similar to Fig. 4, LAMP and ERK tend to preserve parameters, leading to a global parameter distribution that approximates a uniform shape. However, since HARP preserves more parameters in the former layers and the last fully connected layer, more input and output information are preserved.

Figure 10: End-to-end learning processes of weight pruning on a pre-trained robust ResNet18 by using HYDRA (green), HYDRA-ERK (gray), HYDRA-LAMP (blue), and HARP (orange).

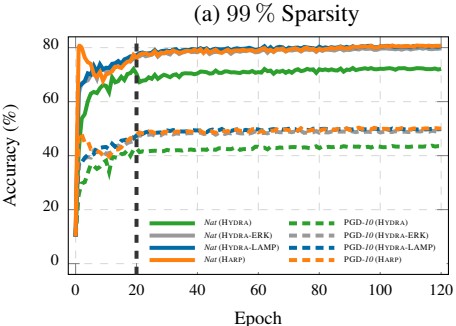
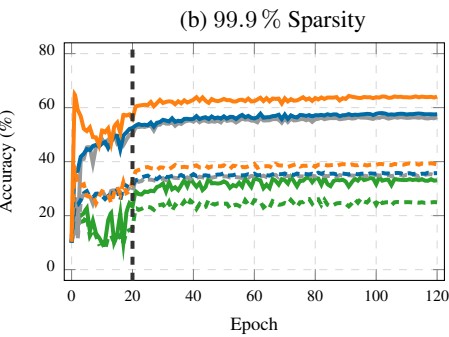

(a) 99 % Sparsity

(b) 99.9 % Sparsity

Table 11: Comparing HARP weight pruning with HYDRA, and HYDRA with ERK and LAMP for ImageNet. Values in brackets show the performance change by using ERK and LAMP on HYDRA.

| Attack | FREE-AT | 90 % Sparsity | | | | 99 % Sparsity | | | |
|---|---|---|---|---|---|---|---|---|---|
| | | HYDRA | | | HARP | HYDRA | | | HARP |
| | | Original | w/ ERK | w/ LAMP | | Original | w/ ERK | w/ LAMP | |
| – | 60.25 | 49.08 | 44.69 (−4.39) | 44.43 (−4.65) | **55.21** | 27.88 | 24.38 (−3.50) | 22.57 (−5.31) | **34.62** |
| PGD | 32.82 | 23.25 | 20.95 (−2.30) | 20.72 (−2.53) | **27.10** | 12.35 | 10.57 (−1.78) | 9.71 (−2.64) | **14.67** |
| C&W$_\infty$ | 30.67 | 21.21 | 19.21 (−2.00) | 19.07 (−2.14) | **24.62** | 11.32 | 9.51 (−1.81) | 8.42 (−2.90) | **12.42** |
| APGD | 31.54 | 22.97 | 20.38 (−2.59) | 20.52 (−2.45) | **25.57** | 12.52 | 10.12 (−2.40) | 9.56 (−2.96) | **13.47** |
| AA | 28.79 | 19.76 | 17.42 (−2.34) | 16.95 (−2.81) | **22.57** | 10.01 | 8.64 (−1.37) | 7.78 (−2.23) | **11.24** |

Figure 11: Strategies of HARP, LAMP, and ERK for pruning a 99 % sparsity on ImageNet.

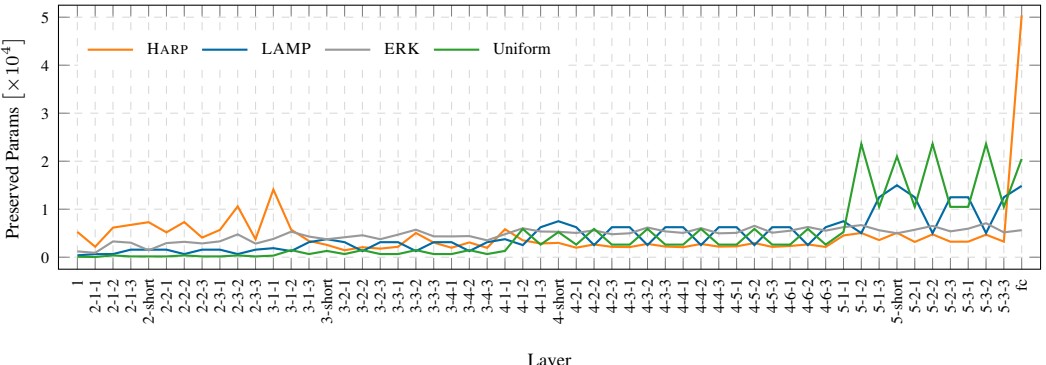

## A.4 PARAMETER DISTRIBUTION

HARP prunes neural networks by incorporating connection importance-scores for learning pruning masks and the layers' compression rates into the optimization procedure. The pruned models particularly benefit from the non-uniformity of HARP's compression strategy, yielding high natural accuracy and high adversarial robustness. Ye et al. (2019) argue that wider parameter distributions promise higher robustness, while Sehwag et al. (2020) show that parameters close to zero are crucial to preserve the adversarial robustness. In this section, we take VGG16 learned on CIFAR-10 and SVHN as examples to analyze the difference in parameter distributions along the network's layers and compare HARP with Robust-ADMM and HYDRA.

**Parameter distribution when pruning weights.** Fig. 12a and Fig. 12b visualize the layer-wise weight distribution of VGG16 pruned by Robust-ADMM (blue), HYDRA (green), and HARP (orange) on datasets CIFAR-10 and SVHN, respectively. Concatenated architectures such as VGG16 possess relatively high sparsity in the middle layers (Lee et al., 2021; Evci et al., 2020). This phenomenon is also captured by HARP, which learns to preserve the first six layers and the last layer.

Figure 12: Parameter distribution of VGG16 pruned to 99 % sparsity with PGD-AT.

(a) CIFAR-10

(b) SVHN

Another intriguing but interesting observation concerns non-zero parameters: Robust-ADMM tends to favor middle layers that contain many zero parameters. In contrast, HYDRA tries to push the model's weights "away" from zero, which reduces sparsity and thus achieves higher robustness than Robust-ADMM. Pruning weights with HARP, however, converges to a different distribution. All but the first and last layers, have weights with (mostly) smaller magnitude than for models pruned with HYDRA—few of them, however, remain zero-valued. For instance, in the penultimate layer, preserved weights are close to zero and the distribution is more compact than those of HYDRA and Robust-ADMM. By comparing the parameter distribution of middle layers in CIFAR-10 and SVHN, we observe that HYDRA tends to have a rather close-zero distribution in pruning networks for SVHN than its behavior in CIFAR-10. Recall Table 2 and Table 3, the performance gap between HYDRA and HARP in SVHN is smaller than in CIFAR-10. This implies an interesting phenomenon that encouraging a close-zero distribution yields a positive impact on pruning robust networks.

## A.5 PERFORMANCE WITH NATURAL TRAINING

We extend our experiments to pruning naturally pre-trained networks with datasets CIFAR-10 and SVHN, and consider HYDRA, HYDRA with LAMP, and the original LAMP pruning in the comparison. In Table 12, there exists no significant difference between the pruning methods at sparsity $90\%$. For CIFAR-10, HYDRA yields a larger performance drop at sparsity $99\%$. In contrast, using LAMP directly or using HYDRA with LAMP's strategy yields better results. With regards to the highest sparsity $99.9\%$, LAMP pruning and HYDRA with LAMP strategy have significantly better performance than original HYDRA. Meanwhile, HARP shows the highest compatibility to prune networks for CIFAR-10. Additionally, in pruning a sparsity $99.9\%$ for SVHN, HARP presents the pruning performance that deviates from the best result by no more than $0.5\%$. Conclusively, pruning naturally pre-trained networks benefits more from the layer-wise specific non-uniform strategies. At the same time, HARP offers more significant improvement about concerning pruning robust networks than other approaches.

Table 12: Comparing pruning weights of naturally pre-trained networks

| Model | | CIFAR-10 | | | | | SVHN | | | | | |
|---|---|---|---|---|---|---|---|---|---|---|---|---|
| | Pre-train | Sparsity | HYDRA | LAMP | HYDRA w/ LAMP | HARP | Pre-train | Sparsity | HYDRA | LAMP | HYDRA w/ LAMP | HARP |
| VGG16 | 93.68 | 90% | 93.29 | **93.82** | 93.79 | 93.59 | 95.71 | 90% | 95.79 | **96.10** | 95.97 | 95.71 |
| | | 99% | 87.25 | 91.49 | 92.86 | **93.01** | | 99% | 95.47 | 95.68 | **95.93** | 95.55 |
| | | 99.9% | 34.62 | 76.52 | 85.61 | **87.42** | | 99.9% | 43.59 | 87.22 | 93.91 | **94.10** |
| ResNet18 | 95.13 | 90% | 94.86 | 95.09 | 95.10 | **95.28** | 96.13 | 90% | 96.23 | 96.32 | **96.36** | 96.10 |
| | | 99% | 83.56 | 93.63 | **94.53** | 94.19 | | 99% | **96.10** | 95.80 | 95.95 | 95.73 |
| | | 99.9% | 25.66 | 78.26 | 86.27 | **86.67** | | 99.9% | 70.78 | 90.67 | **93.79** | 93.32 |

## A.6 EXTENSION TO CHANNEL PRUNING

Structurally pruning neural networks has better compatibility with the hardware deployment, while its coarse pruning granularity leads to larger performance degradation than weight pruning, itself increasing the difficulty of pruning robust networks. In this section, we extend our method HARP on pruning network layer channels. To map HARP on channel pruning, we use FLOPs estimation (denoted as function $FLOPS(\cdot)$) to replace the sparsity in $\mathcal{L}_{hw}$ (expressed as Eq. (8)). Moreover, we determine $a_t$ by considering network global FLOPs reduction, in order for the pruned model to have the same size as a uniform strategy would have in FLOPs.

$$\mathcal{L}_{hw}(\hat{\boldsymbol{\theta}}, a_t) := \max \left\{ \frac{FLOPs\left(\hat{\boldsymbol{\theta}}\right)}{a_t \cdot FLOPs\left(\boldsymbol{\theta}\right)} - 1 \, , \, 0 \right\} \tag{8}$$

Different from weight pruning, importance scores $\boldsymbol{S}^{(l)}$ originates $\mathbb{R}^{c_i^{(l)}}$ for channel pruning. In addition, we use Eq. (9) to initialize $\boldsymbol{S}^{(l)}$ in channel pruning and follow the order of weight magnitude (Ye et al., 2019) by summing up the weights of each input channel (expressed as $\mathrm{csum}(\cdot)$) and normalizing by the maximal channel-wise summation.

$$\boldsymbol{S}_C^{(l)} = \left( \eta \cdot \frac{\text{csum}(|\boldsymbol{\theta}^{(l)}|)}{\max(\text{csum}(|\boldsymbol{\theta}^{(l)}|))} \right)_{c_i^{(l)}} \tag{9}$$

Algorithm 1 presents the complete implementation of HARP. Note that weight pruning is largely independent of the network architecture as it does not change the layer's size (we merely zero out weights). However, pruning channels/filters requires special attention. When facing ResNet-like architectures, we do not train the compression rates of all shortcut layers directly but update them by assigning the rate of the connected input layer in the residual block instead. This way, the pruned input layer aligns with the channel dimensionality of connected pruned shortcut-layers.

**Comparing channel pruning with related work.** Ye et al. (2019) have shown the capability of Robust-ADMM in structurally pruning robust networks, controlling the network FLOPs straightforwardly by a global structural compression rate. Similarly, HYDRA (Sehwag et al., 2020) extends to structural pruning and improves over magnitude-based criterion (Han et al., 2015). For channel pruning with HARP, we start off from the uniform strategy with $a_{init} = 1.0$ and use $\gamma = 0.02$ to ensure the arrival at the target compression rate $a_t$. As presented experiment results in Fig. 13, HARP shows promising performance in preserving a robust model while pruning channels. Robust-ADMM possesses a similar performance on pruning up to $4.0$ xFLOPs, while experiencing higher model performance degradation at xFLOPs of 10 and 20. Channel pruning with HYDRA, in turn, significantly harms model performance at $2.0$ xFLOPs compression.

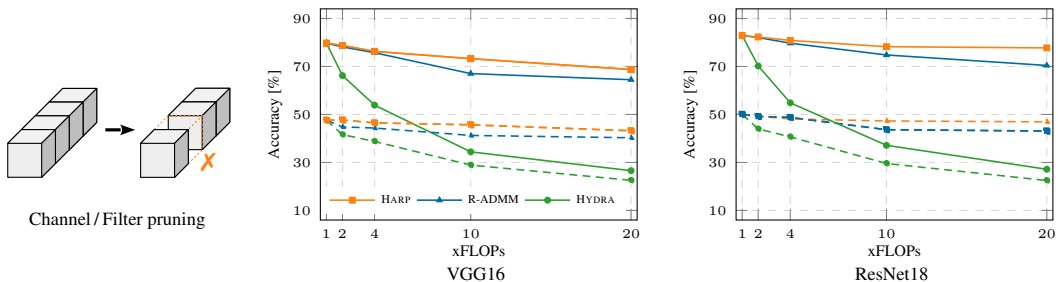

Figure 13: Structural pruning by controlling FLOPs on a VGG16 model and a ResNet18 model for CIFAR-10. Solid lines show the natural accuracy of HARP (orange), HYDRA (green) and Robust-ADMM (blue). Dashed lines represent the methods' adversarial robustness of PGD-10.

**Comparison to ANP-VS.** Madaan et al. (2020) propose to suppress the latent feature level vulnerability in neural networks by gradually pruning a model during training based on a regularized loss function. In the open-source implementation of ANP-VS [1], pre-processing CIFAR-10 data is done using image standardization (e.g., `tf.image.per_image_standardization()` function). Standardization transforms the input to have mean $0.0$ and variance $1.0$, that is, input features may fall outside of an $[0.0, 1.0]$ interval. However, ANP-VS clips adversarial perturbations to $[0.0, 1.0]$, resulting in overall weaker adversarial examples used for the evaluation as it would be the case for 0-1 rescaling the inputs. Note, the latter is used for evaluating HARP in all experiments. We investigate the influence of input ranges by comparing training with ANP-VS using both pre-processing steps (standardization and simple 0-1 re-scaling) under the otherwise exact same PGD settings. The original publication reports an original accuracy of $88.18\,\%$ adversarial accuracy of $56.21\,\%$ for VGG16 on CIFAR-10, which match our reproduced results with standardization (cf. Table 13). We even yield higher compression in our experiments. With the correct input pre-processing step (0-1 rescaling),

Table 13: Comparing ANP-VS with different pre-processing methods.

| Pre-processing | xFLOPs | Benign | PGD-40 |
|---|---|---|---|
| Original (Madaan et al., 2020) | 2.41 | 88.18 | 56.21 |
| Standardization | 3.56 | 88.34 | 57.33 |
| 0-1 re-scaling | 3.82 | 76.30 | 31.10 |

however, ANP-VS yields a significantly worse result, showing the criticality of wrongly used pre-processing. With thus refrain from comparing HARP to ANP-VS in pruning channels.

---

[1] https://github.com/divyam3897/ANP_VS

---

**Algorithm 1** Holistic Adversarially Robust Pruning

---

**Input:** pre-trained model and its parameters $\boldsymbol{\theta}$, number layers $L$, target compression rate $a_t$, minimal compression rate $a_{min}$, training data-set $\mathcal{D}$

**Output:** Compressed network $\widetilde{\boldsymbol{\theta}} \odot \boldsymbol{M}^*$

**Step 1:** Initialization of trainable pruning parameters

For each layer $l$, initialize learnable rates $^{(l)} = \log\left(\dfrac{a_{init} - a_{min}}{1 - a_{init}}\right)$

and importance scores:

$$\boldsymbol{S}_W^{(l)} = \left(\eta \cdot \frac{\boldsymbol{\theta}^{(l)}}{\max(|\boldsymbol{\theta}^{(l)}|)}\right)_{c_i^{(l)} \times c_o^{(l)} \times k^{(l)} \times k^{(l)}} \qquad \text{or} \qquad \boldsymbol{S}_C^{(l)} = \left(\eta \cdot \frac{\text{csum}(|\boldsymbol{\theta}^{(l)}|)}{\max(\text{csum}(|\boldsymbol{\theta}^{(l)}|))}\right)_{c_i^{(l)}}$$

**Step 2:** Strategy search via robust training

$$\boldsymbol{M}^* = \arg\min_{\boldsymbol{M}} \mathop{\mathbb{E}}_{(\boldsymbol{x},y)\sim\mathcal{D}} \left[\max_{\delta} \left\{\mathcal{L}_{robust}(\boldsymbol{\theta} \odot \boldsymbol{M}, \boldsymbol{x} + \delta, y)\right\}\right] + \gamma \cdot \mathcal{L}_{hw}(\boldsymbol{\theta} \odot \boldsymbol{M}, a_t),$$

with $\boldsymbol{M} = (\boldsymbol{M}^{(1)}, \dots, \boldsymbol{M}^{(l)}, \dots, \boldsymbol{M}^{(L)})$ and $\boldsymbol{M}^{(l)} := \left(\mathbb{1}_{s > P(\alpha^{(l)}, \boldsymbol{S}^{(l)})}\right)$

**Step 3:** Fine-tuning of the pruned network $\hat{\boldsymbol{\theta}} = \boldsymbol{\theta} \odot \boldsymbol{M}^*$

$$\widetilde{\boldsymbol{\theta}} = \arg\min_{\hat{\boldsymbol{\theta}}} \mathop{\mathbb{E}}_{(\boldsymbol{x},y)\sim\mathcal{D}} \left[\max_{\delta} \left\{\mathcal{L}_{ce}(\hat{\boldsymbol{\theta}}, \boldsymbol{x} + \delta, y)\right\}\right]$$

---

## A.7 Training Consumption

In this section, we elaborate on the training (time) consumption of the pruning approaches considered in our evaluation. As every method operates on the complete training dataset, we estimate the consumption in training epochs rather than wall-clock time as the latter varies with the used hardware and its load. We report the results in Table 14. The methods either follow a three-stage pipeline (Han et al., 2015) or train the model from scratch with pruning considerations in their optimization. Consequently, the distribution of training effort per stage varies. Under this metric, MAD is most efficient. However, when targeting a sparsity of $99\,\%$, HARP arrives close its to best performance right after pruning (20 epochs) as can be seen in Fig. 10. At $99.9\,\%$ sparsity, our method uses 100 epochs for fine-tuning to recover natural accuracy but does not improve robustness anymore. Thus, HARP's pruning stage turns out to be most decisive but helps for aggressive pruning.

Table 14: Training consumption in different robust pruning methods

| Method | Multi-Stage Pipeline (Han et al., 2015) | Number of Epochs | | | |
| --- | --- | --- | --- | --- | --- |
| | | Pre-training | Pruning | Fine-tuning | Total |
| R-ADMM (Ye et al., 2019) | ✓ | 100 | 100 | 100 | 300 |
| HYDRA (Sehwag et al., 2020) | ✓ | 100 | 20 | 100 | 220 |
| BCS-P Özdenizci & Legenstein (2021) | ✗ | — | — | — | 200 |
| DNR (Kundu et al., 2021) | ✗ | — | — | — | 200 |
| MAD (Lee et al., 2022) | ✓ | 60 | 20 | 60 | 140 |
| **HARP (Ours)** | ✓ | 100 | 20 | 100 | 220 |

## A.8 CO$_2$ Emission

We have conducted all our experiments on Nvidia RTX-3090 GPU cards and have consumed about 10,198 GPU hours in total. This amounts to an estimated total CO$_2$ emissions of 2,177.27 kgCO2eq when using Google Cloud Platform in region `europe-west3`. However, our university consumes $100\,\%$ renewable-energy, such that our specific CO$_2$ emissions for the project amounts to 1.74 kgCO2eq. Estimates are conducted using the "Machine Learning Impact Calculator" (Lacoste et al., 2019).

