# OpenReview forum: "Holistic Adversarially Robust Pruning"
_ICLR.cc/2023/Conference — ICLR 2023 poster_

### Official Review · Reviewer_LsE9 · 2022-10-20

**Confidence:** 5
**Correctness:** 3
**Technical Novelty And Significance:** 2
**Empirical Novelty And Significance:** 3
**Recommendation:** 8

**Clarity, Quality, Novelty And Reproducibility:**

Clarity: The paper is written in easily understandable fashion. Nevertheless, there should be more details/clarifications with regards to the training and attack evaluations settings in the manuscript (e.g., in table/figure captions).

Quality & Novelty: The proposed approach appears to be motivated by the score optimization based pruning method of HYDRA, combined with the layer-wise non-uniform pruning capability of BCS-P which was shown to outperform HYDRA. While methodologically incremental in that sense, the algorithm appears to yield significant results.

Reproducibility: Sufficient, since the authors provided their code and resulting evaluation outputs in the supplement. However some of the documented results that correspond to BCS-P and MAD comparisons do not really contain the evaluations presented in Tables 2 & 3, especially the evaluations at very high sparsity 99.9%.



**Strength And Weaknesses:**

Strengths: The paper is well written and easy to understand. Proposed algorithm scales to ImageNet and outperforms some previous methods at that scale. It is also nice that the approach is compatible with any robust training objective similar to the line of work by HYDRA and BCS-P.

Weaknesses: Details describing adversarial robustness evaluation settings should be also present in the main manuscript rather than only the Supplementary. Current experimental comparisons need to be presented in a more rigorous fashion and also some of the evaluations at very high sparsity levels needs further analysis/clarification (see comments below).




**Summary Of The Paper:**

The paper proposes an algorithm for pruning of robust neural networks optimized under adversarial training objectives. The algorithm relies on simulatenously optimizing the (per-layer) compression rates and importance scores corresponding to the connections  that can be pruned, within a dynamic regularization scheme that balances the two objectives. This way the method allows a layer-wise non-uniform pruning scheme that preserves a whole-network sparsity rate at the end of optimization. Experimental results on CIFAR-10, SVHN and ImageNet show that the approach yields benefits particularly at very high sparsity rates starting from 99% and above.


**Summary Of The Review:**

Generally I am convinced that the algorithm is neat and it indeed appears to be more powerful than R-ADMM (2019) and HYDRA (2020) type of methods that are more restricted to layer-wise uniform pruning percentages. However there are some ambiguities that needs to be clarified and I list my points below:
- Overall, all figure comparisons should be also performed with a stronger attack than PGD-10 (which can be highly misleading here). Assuming that models were trained with 10-step PGD for any robust training loss as well, attack evaluations should be at least presented with 20- or 50-step PGD with 5 or 10 restarts. Going further, since the authors have involved AutoAttack evaluations, one could keep all evaluations simple by presenting all results in that setting too. For instance, can the authors present robust accuracies in Figures 1 2 and 3 under AA?
- Attack configurations are major details that should be part of the main manuscript (at least summarized). Looking at the figure and table captions/legends, it is not possible to see with which objective the models were trained with (e.g., how many PGD steps), and what is the evaluation setting of the attack (i.e., equal number of PGD steps used during training?). For instance in Figures 2 and 3, robust training loss and attack evaluation settings are not clearly annotated.
- Authors already state in their paper that BCS-P (2021) previously proposed a non-uniform compression strategy that outperformed HYDRA and R-ADMM. This already makes BCS-P a more relevant comparison baseline for HARP to outperform at all categories, but those evaluations appear shallow (only a single VGG16 model under the TRADES objective is studied). Can the authors extend these results on page 7 with a e.g., ResNet-18 and other robust training objectives? In that vein, for instance, authors can also include BCS-P in the plots of Figure 1.
- The provided ImageNet model checkpoints by the authors that correspond to the Table 5 of the HYDRA paper, do not match with the results in the authors submission (Table 3). It appears like using the same dense model with Free-AT (60.25 clean acc), HYDRA achieves a different model at the end for e.g., 99% sparsity? Can the authors verify/discuss this?
- All results compared at 99.9% appear like other methods could not get to work at that level of sparsity at all (i.e., trainings are early terminated as all clean accuracies are at chance level). At the right halves of Table 1 and 2 all accuracies are ~10-30%. How much in depth did the authors explore these methods (such best attempt details can be described in the Supplementary)? Similarly in Figure 2a and b, BCS-P appears to yield a randomly initialized 10% performing network. How much did the authors explore this method?
- It appears like HARP necessitates a dense model to be adversarially trained in the first place, and moves on to a compute-heavy adversarial pruning phase afterwards. Given these limitations, a brief discussion of all compared methods in terms of training time and computational cost would be necessary, until one achieves the final sparse model with an associated clean/robust accuracy.
- Can the authors provide a simulation/visualization of (possibly occurring) robust overfitting during the final stages of pruning at very high sparsity, while adapting gamma?
- Figure 3 caption has a typo, HYDRA -> HARP. Also I believe comparisons to MAD would perhaps look better in a short table, since the available data points are very sparse.
- The x-axis scale in Figure 1 plots looks a bit off?
- There is a figure in page 4 without label/caption/number.

---

> ### Author Response · Authors · 2022-11-14
> **Reply to reviewer LsE9 (2)**
>
> ### Ques-5: Analysis on model collapse at 99.9%
>
> We explore the model collapse of related work based on the layer-wise distribution of their strategies. In the revision, we have added BCS-P pruning a VGG16 at a sparsity of 99.9% in Fig. 4. Interestingly, BCS-P training results in an almost uniform pruning strategy. Consequently, the first layer is pruned by 99.9%. However, VGG16's first layer has 1,728 parameters, and 0.1% preserved parameters refer to merely $2$ parameters. This, of course, is way less than needed for good model performance.
>
> ### Ques-6: Discussion on training consumption
>
> Pruning and fine-tuning are based on adversarial training using the PGD attack on the complete training dataset for all methods. To compare the training costs irrespective of the underlying hardware differences (GPU model and load), we report the number of training epochs that achieve the best model robustness in the tables below. Each epoch is run on the complete training dataset.
>
> Table (a): Training epochs for different robust pruning methods in pruning VGG16 weights on CIFAR10.
>
> | Sparsity | Method   | Pretrain  | Pruning   | Fine-tuning  | Total $\downarrow$   |
> |----------|----------|-----------|-----------|--------------|----------------------|
> |    90%   | R-ADMM   | 100       | 100       | 2	     | 144                  |
> |          | Hydra    | 100       | 20        | 1            | **121**              |
> |          | BCS-P    | ---       | ---       | ---          | 199                  |
> |          | HARP     | 100       | 20        | 1            | **121**              |
> |    99%   | R-ADMM   | 100       | 100       | 98	     | 298                  |
> |          | Hydra    | 100       | 20        | 84           | 204                  |
> |          | BCS-P    | ---       | ---       | ---          | 187                  |
> |          | HARP     | 100       | 20        | 38           | **158**              |
>
> Table (b): VGG16's performance on CIFAR10 after pruning and fine-tuning by methods that follow three-stage pipeline.
>
> | Sparsity | Method   | Only Pruning   | After Fine-tuning | Delta $\downarrow$ |
> |----------|----------|----------------|-------------------|--------------------|
> |    90%   | R-ADMM   | 68.64 / 38.13  | 75.41 / 46.50     | +6.77 / +8.37     |
> |          | Hydra    | 77.00 / 45.97  | 77.31 / 47.86     | +0.31 / +1.89      |
> |          | HARP     | **77.62** / **47.86**  | **78.18** / **48.59**     | +0.76 / +0.73      |
> |    99%   | R-ADMM   | 25.92 / 16.41  | 61.84 / 36.83     | +35.92 / +20.42    |
> |          | Hydra    | 54.32 / 34.85  | 65.09 / 39.80     | +25.29 / +4.95     |
> |          | HARP     | **76.04** / **46.68**  | **77.89** / **48.51**     | +1.85 / +1.83      |
>
> HARP consumes the least number of epochs to find the best model after pruning. Moreover, Table (b) that also HARP's pruning phase is the most effective. In our revision, Fig. 10 shows that HARP does not require the full budget for reaching the best robustness. In particular, our method reach the maximum after the first epoch of fine-tuning for a sparsity of 90% already. However, for high compression/sparsity our method makes good use of the granted epochs.
>
> ### Ques-7: Robustness overfitting by adapting gamma
>
> In Appendix A.1 of the revised version of our paper, we present an analysis of the step size's influence on HARP. In Fig. 5, we show the learning progress for different values of $\gamma$.
> For $\gamma > 0.1$, HARP oscillates between model compression and the natural/adversarial performance. In Fig. 5(c), we see that high values of $\gamma$ allow arriving at the target compression rate after the first few epochs. However, for $\gamma=0.01$, the oscillation starts and gets stronger with decreasing step sizes. $\gamma=0.001$, in turn, fails to yield the target compression.
>
> ### Ques-8: Typo + MAD results as table
>
> Thank you very much for the suggestions. We have factored out the results for 90% sparsity to Table 5 and included HYDRA & R-ADMM in Fig. 3 and Fig. 8, showing the robust accuracy of MAD and HARP.
>
> ### Ques-9: X-axis scale is off in Fig. 1
>
> Indeed, the presentation in the original submission was not ideal. The revised submission now contains figures on a logarithmic scale to highlight high sparsity. Please refer to Figs. 1, 3, 6, and 8.
>
> ### Ques-10: Wrapfigure without caption
>
> Thank you, we have adjusted the formatting accordingly.

---

> > ### Comment · Reviewer_LsE9 · 2022-11-18
> > **Thanks to the authors for their responses and clarifications**
> >
> > I think the authors did a great job in the rebuttal and I appreciate their effort. Most of the ambiguities are cleared on my side. Robustness evaluations look more rigorous and (importantly) consistent now. Extended HYDRA and BCS-P experiments appear solid. I don't have further questions, as my main concern on the model collapses at 99.9% sparsity is somewhat fairly clarified in the manuscript. I increase my score in response to the authors' efforts.
> >
> > One minor comment: The paper includes many abbreviations, and most of them are really undefined until quite late. Perhaps authors should define these abbrv. names earlier (e.g., BCS-P, MAD, ADMM, ERK, etc.).

---

> > > ### Author Response · Authors · 2022-11-18
> > > **Reply to reviewer's response**
> > >
> > > We sincerely thank the reviewer for the positive feedback and the adjusted score. For the final submission, we will ensure that all abbreviations are introduced and referenced as soon as they are used in the text.

---

> ### Author Response · Authors · 2022-11-14
> **Reply to reviewer LsE9 (1)**
>
> We thank the reviewer for the feedback and constructive suggestions. We provide answers to the raised questions below and are looking forward to your response to be able to make the appropriate changes to the paper.
>
> ### Ques-1: Replacing robust eval with AA
>
> In Fig. 1, Fig. 3, and Fig. 6, we now evaluate against AutoAttack, as also reported in Tables 2 and 3. While the absolute values change, of course, the tendency of each method remains unchanged. HARP still outperforms other methods with increasing sparsity.
>
>
> ### Ques-2: Lack of evaluation setting in main part
>
> Thank you for the suggestion. We have included descriptions of the adversarial training and evaluation settings in Section 4 of the main part.
>
>
> ### Ques-3: BCS-P extension and investigation
>
> We include the results of BCS-P with PGD-10 training in Fig. 1. In Fig. 6 of the appendix, we now show results for comparing HARP and BCS-P on ResNet18. Both approaches are also evaluated on SVHN (Fig. 1 and 6). Table 7 summarizes the complete comparison with three different training methods.
> Moreover, we have investigated the reasons for BCS-P's poor performance at a sparsity of 99.9%, which can best be seen in Fig.4 and Fig.7, showing the found strategy and the parameter distribution. Interestingly, BCS-P yields an almost uniform strategy, although it strives for non-uniformity. Even worse, only two out of 1,728 parameters (i.e., 64x3x3x3) in VGG16's input layer are preserved at 99.9 % sparsity, severely impacting performance.
>
> ### Ques-4: Biased Hydra ImageNet results:
>
> We can confirm the reviewer's observation.
>
> In our experiments for ImageNet, we use the pre-trained models offered by the authors of HYDRA. The checkpoints in their GitHub repository indeed fall behind the performance reported in the paper. The table below lists the specific values. Please, note that PGD-10 accuracy might slightly differ due to its intrinsic randomness.
>
> | Sparsity | Evaluation | Results in Paper | Checkpoints |
> |----------|------------|------------------|-------------|
> | 0%       | Clean      | 60.2%            | 60.25%      |
> |          | PGD-10     | 32.0%            | 32.95%      |
> | 95%      | Clean      | 47.1%            | 44.59%      |
> |          | PGD-10     | 21.4%            | 20.11%      |
> | 99%      | Clean      | 31.5%            | 27.66%      |
> |          | PGD-10     | 13.0%            | 11.80%      |
>
>
> Additionally, we re-run HYDRA with the offered pre-trained robust ResNet50 model on ImageNet to investigate the issue's extent. Please refer to the log files in the supplementary material:
>
> * Folder 'supplementary/hydra_checkpoint_eval/'.

---

### Official Review · Reviewer_GDVf · 2022-10-22

**Confidence:** 4
**Correctness:** 3
**Technical Novelty And Significance:** 2
**Empirical Novelty And Significance:** 3
**Recommendation:** 6

**Clarity, Quality, Novelty And Reproducibility:**

The paper is easy to follow. The technical contribution is marginal; the empirical results are surprisingly good.

**Strength And Weaknesses:**

Strength
- It is good to see that the non-uniform compression strategy is beneficial also to adversarial (channel) pruning.
- It shows large improvement over prior methods, robust-ADMM, HYDRA, BCS-P, and MAD.
- It contains comprehensive discussion, extensive comparisons and ablation study.

Comments
- As structural pruning is more useful and harder than weight pruning, I would suggest authors move some channel pruning results to the main text and compare with SOTAs.
- It would be better to add the pre-trained model's performance to both Table-1 and Table-2.
- In Table-2, why the variances of HYDRA with TRADES under 99% sparsity are such high? Can authors provide some insights about it?
- Will the authors release the code?



**Summary Of The Paper:**

This paper proposes a new method to prune a robust DNN with layer-wise compression rate and score-based pruning masks. Experimental study shows that, by taking advantage of both two factors, it maintains accuracy while allows for less robustness degradation than SOTAs.

**Summary Of The Review:**

This paper proposes a method to prune a robust DNN with a non-uniform compression strategy. Interestingly the proposed pruning method outperforms existing ones both in terms of robustness and accuracy.

---

> ### Author Response · Authors · 2022-11-14
> **Reply to reviewer GDVf**
>
> We appreciate the reviewer's comment on our method's effectiveness and thank the reviewer for the feedback. Below, we provide the answers to the raised concerns. We are looking forward to your response to be able to make the appropriate changes to the paper.
>
> ### Ques-1: Structural pruning with SoTAs
>
> Thank you for the kind suggestion. Most related work focuses on weight pruning, and HARP can contribute most to this strain of research. We thus want to keep the current focus and maintain our experiments on structural pruning as a complementary analysis.
>
> We leave a more extensive consideration of structural pruning to future work.
>
>
> ### Ques-2: Having pretrained model in the main part
>
> We agree with the reviewer's suggestion and have merged the results of our pre-trained models into Tables 1 and 2 of our revised paper.
>
> ### Ques-3: High variance at 99.9% in Hydra TRADES
>
> Good catch! Thank you very much for pointing this out.
>
> Upon careful investigation, we found out that one single run had been interrupted prematurely and biased the overall results (and variance) for our experiments on pruning ResNet18 at 99% sparsity with HYDRA. The updated results in Table 2 show that HYDRA improves both in overall robustness and variance compared to the initial submission. However, at high sparsity (99.9%), HARP remains superior.
>
>
> ### Ques-4: Code release
>
> Glad you asked. For reproducibility and to foster future research, we will make all code and intermediate results publicly available in a GitHub repository. A snapshot of our implementations is also available in the supplementary material (folder `harp-prune/`).
>
> [1] Adversarial Robustness vs. Model Compression, or Both? ICCV 2019.
> [2] HYDRA: Pruning Adversarially Robust Neural Networks, NeurIPS 2020.
> [3] Adversarial Neural Pruning with Latent Vulnerability Suppression, ICML 2020.

---

### Official Review · Reviewer_j8jC · 2022-10-24

**Confidence:** 5
**Correctness:** 3
**Technical Novelty And Significance:** 2
**Empirical Novelty And Significance:** 2
**Recommendation:** 3

**Clarity, Quality, Novelty And Reproducibility:**

Clarity: 7/10

Quality: 5/10

Novelty: 3/10

Reproducibility: 8/10.


**Details Of Ethics Concerns:**

N/A.

**Strength And Weaknesses:**

## Strengths
=============
1. The paper is well written and organized.
2. Results are inspiring and thorough. In particular, detailed results on large scale datasets like ImageNet is good.
3. Literature survey is well up to date apart from few missing ones.

## Weakness
==============
1. Please provide comparison with DNR [1] method. DNR provides better compression-vs-accuracy trade-offs for smaller and less deeper models like ResNet18, VGG16 (as per robustbench: https://github.com/RobustBench/robustbench).

2. The paper uses pre-trained models to start compression and adversarial fine tuning, while there are existing works that did it from scratch [1].

3. The contribution section is not clear, the authors mention about experiments in the contribution. Please be more clear in stating what is the explicit contribution of this manuscript.

4. The question ***How many and which parameters to prune***, is not exactly correct in my opinion, as the ***how many*** part is governed by the target hardware's parameter support. Also, again, the learning of layer-wise pruning, that is indeed a crucial part of robust compression is being handled earlier [1].

5. Connection importance score has been well studied as well in the pruning literature [2-3].

Based on the above four facts, I would really recommend the authors to reconsider the motivation of the current manuscript. This also translates to lack of significant contribution for the paper to cross the border of ICLR acceptance.

6. Please detail about the training cost for HARP, including that of pre-training, to reach a specific parameter budget.

7. The robust compression details are not clear. In particular, whether the authors use traditional PGD-7 attack to create adversary or leverage faster variants like FAT [4], should be mentioned with more clarity.

8. Also, HYDRA is not the SOTA paper on robust pruning starting from a pretrained model.

[1] DNR: A Tunable Robust Pruning Framework Through Dynamic Network Rewiring of DNNs, ASP-DAC 2021.

[2] Snip: Single-shot network pruning based on connection sensitivity, ICLR 2019.

[3] Pruning neural networks at initialization: Why are we missing the mark?, ICLR 2021.

[4] Fast is better than free: Revisiting adversarial training, ICLR 2020.

**Summary Of The Paper:**

The paper presents a unified model compression to improve model robustness by learning the connection importance for each layer weights. In particular, the authors use a pre-trained model to learn the layer importance to meet a specific pruning budget, and gradually increases the pruning % unless a significant accuracy drop is observed.

**Summary Of The Review:**

The paper presents a unified robust pruning framework that can reach SOTA accuracy by honoring the layer-wise connection sensitivity.

---

> ### Author Response · Authors · 2022-11-14
> **Reply to reviewer j8jC (2)**
>
>
> ### Question 3. Unclear contributions
>
> We have restructured the paragraph of contributions in Section 1, "Introduction" of our paper and to list three key contributions:
>
> 1. We propose a novel pruning technique for pre-trained models based on optimizing layer-wise compression and scoring connections.
>
> 2. We significantly improve over related work and demonstrate this in extensive experiments across datasets, adversarial training methods, and models.
>
> 3. We empirically show the importance of non-uniform pruning for adversarial robustness by evaluating novel extensions of HYDRA and R-ADMM to non-uniform compression.
>
> Please find details at the end of the paper's introduction.
>
> ### Question 4.
>
> As the reviewer correctly observes, the overall compression rate is, of course, governed by the hardware's limitations. Our approach determines how many parameters to prune /per layer individually/ to yield (a) high robustness and (b) the requested overall compression rate.
> We have clarified this subtle difference in the Abstract, the list of contributions in the introduction, and Section 3, describing our method.
>
> ### Question 5.
>
> In prior work, connection importance scores have been used in traditional pruning [2,3] and for adversarially robust pruning [10]. We explicitly acknowledge this advancement and build upon the community's insights to design HARP. We extend this line of research by additionally incorporating optimization of compression rates /by layer/ to arrive at the network's overall compression rate. We clarify this dual objective in the Abstract, the contributions, and Section 3.
>
> ### Question 6.
>
> All methods are applied to the complete training dataset. For comparing the training costs irrespective of the underlying hardware differences (GPU model and load), we report the number of training epochs below. Each epoch is run on the complete training dataset.
>
>
> | Method   | From Scratch |Pre-train  | Pruning  | Fine-Tuning | Total |
> |----------|--------------|-----------|----------|-------------|-------|
> | R-ADMM   | No           | 100       | 100      | 100         | 300   |
> | Hydra    | No           | 100       | 20       | 100         | 220   |
> | BCS-P    | Yes          | ---       | ---      | ---         | 200   |
> | MAD      | No           | 60        | 20       | 60          | 140   |
> | HARP     | No           | 100       | 20       | 100         | 220   |
>
>
> These numbers are reported in Appendix A.7 in the revised submission.
>
>
> ### Question 7.
>
> We have updated our description of the adversarial training settings in Section 4 of the revised paper. In a nutshell, PGD-AT, TRADES-AT, and MART-AT use a perturbation budget of $\epsilon=8/255$ with step size $\alpha=2/255$ for $10$ steps. For ImageNet, we use Free-AdvTrain [5] with a perturbation budget of $\epsilon=4/255$, a step size $\alpha=1/255$, and $10$ steps to accelerate adversarial training. The number of repeats in Free-AdvTrain is set to $4$.
>
> ### Question 8.
>
> HYDRA [10] poses a landmark in adversarially robust pruning; thus, we consider it a worthwhile baseline. However, as the reviewer correctly points out, HYDRA is not state-of-the-art. Consequently, we have extended our evaluation to MAD [7], published at CVPR 2022.
>
> HARP outperforms MAD under aggressive pruning. At a sparsity of 99.9%, HARP achieves similar adversarial robustness as MAD at a sparsity of 99%, meaning the network pruned by MAD is an order of magnitude larger.
>
> ---
>
>  [1] DNR: A Tunable Robust Pruning Framework Through Dynamic Network Rewiring of DNNs, ASP-DAC 2021
>  [2] Snip: Single-shot network pruning based on connection sensitivity, ICLR 2019.
>  [3] Pruning neural networks at initialization: Why are we missing the mark?, ICLR 2021.
>  [4] Fast is better than free: Revisiting adversarial training, ICLR 2020.
>  [5] Adversarial Training for Free! NeurIPS 2019.
>  [6] Training adversarially robust sparse networks via bayesian connectivity sampling, ICML 2021.
>  [7] Masking adversarial damage: Finding adversarial saliency for robust and sparse network, CVPR 2022.
>  [8] Parametric noise injection: Trainable randomness to improve deep neural network robustness against adversarial attack, CVPR 2019.
>  [9] Learning both weights and connections for efficient neural network, NeurIPS 2015.
> [10] HYDRA: Pruning Adversarially Robust Neural Networks, NeurIPS 2020.

---

> ### Author Response · Authors · 2022-11-14
> **Reply to reviewer j8jC (1)**
>
> We thank the reviewer for the feedback and suggestions. In following, we provide answers to the raised questions below and are looking forward to your response to be able to make the appropriate changes to the paper.
>
> ### Questions 1-2
>
> For evaluating HARP's performance, we first compare with methods following Han et al.'s pretraining-pruning-finetuning pipeline [9] (R-ADMM and HYDRA) and extend to state-of-the-art methods that use a different structure, such as BCS-P [6] and MAD [7]. BCS-P, for instance, yields sparse models from scratch, meaning during the training process itself. HARP improves over BCS-P, as shown in Fig. 1, Fig. 6, and Table 7.
>
> We follow the reviewer's suggestion and compare our method to DNR [1]. Results are provided in Appendix A.2 Table 8. We use the same setting as proposed by the authors of DNR [8] and map the compression rates used in their publication to our sparsity measures.
> DNR favors natural accuracy over robustness, while our method, HARP, tends to maintain higher robustness at the same compression as DNR. When increasing the compression (50x compression -> 98% sparsity), HARP still preserves robustness but loses some natural accuracy.
> Moreover, we run DNR on Tiny-ImageNet and see that HARP yields significantly higher robustness than DNR at the same compression.
>
> | Dataset       | Model    | Method        | Compression       | Sparsity | Clean   | FGSM    | PGD-7    |
> |---------------|----------|---------------|-------------------|----------|---------|---------|----------|
> | CIFAR 10      | VGG16    | DNR           | 21.57x            | 95.4%    |**86.74**| 52.92   | 43.21    |
> |               |          | HARP          | 21.57x            | 95.4%    | 85.09   |**53.55**|**45.34** |
> |               |          | HARP          | **50.00$\times$** |**98.0%** | 83.16   | 52.27   | 45.13    |
> |               |          | HARP          | **100.0$\times$** |**99.0%** | 81.35   | 50.83   | 42.75    |
> |               | ResNet18 | DNR           | 20.85$\times$     | 95.2%    |**87.32**| 55.13   | 47.35    |
> |               |          | HARP          | 20.85$\times$     | 95.2%    | 86.97   |**56.33**|**49.31** |
> |               |          | HARP          | **50.00$\times$** |**98.0%** | 84.30   | 53.87   | 46.78    |
> |               |          | HARP          | **100.0$\times$** |**99.0%** | 82.35   | 52.06   | 45.47    |
> | Tiny-ImageNet | VGG16    | DNR           | 20.63$\times$     | 95.2%    |**51.71**| 18.21   | 14.46    |
> |               |          | HARP          | 20.63$\times$     | 95.2%    | 50.43   |**20.06**|**16.25** |

---

> ### Comment · Reviewer_j8jC · 2022-11-22
> **Acknowledgment of rebuttal**
>
> Dear authors,
>
> I appreciate your effort in providing additional results. However, the strength of the results has diminished a lot. In particular, DNR that does not require any pre-trained model, can get close to or provide better results than HARP( on clean images). Due to additional compute burden of iterative training this does not seem like a strong result anymore. Moreover, I see this paper using existing tricks to re invent some old wheels, for example layer wise allocation of parameters based on suitable sensitivity. Thus, despite the results being thorough I can't vote for acceptance.

---

> > ### Author Response · Authors · 2022-11-22
> > **Reply to further response**
> >
> > We kindly disagree with the reviewer's conclusion and want to point out the fundamental difference between HARP and DNR:
> >
> > - HARP owns the control to yield a specific target compression rate, but DNR is not.
> > - HARP excels for high compression rates, while DNR only yields moderate compression.
> >
> > The approach closest to our objective, shared by HYDRA (Sehwag et al., NeurIPS 2020) and MAD (Lee et al., CVPR 2022), but uses training from scratch is BCS-P (Özdenizci and Legenstein, ICML 2021). We can significantly outperform BCS-P, as shown in Fig. 1 and Appendix A.2.

---

> > > ### Comment · Reviewer_LsE9 · 2022-11-22
> > > **agreeing with the authors**
> > >
> > > I agree with this perspective of the authors actually. I'll join in to help on emphasizing this for Reviewer j8jC.
> > >
> > > In my opinion, comparisons to DNR are not hurting this paper's position much. DNR is not an algorithm where one can achieve extreme sparsity levels of around 99.9%. Going further, DNR does not also allow compressing a network with a user defined sparsity level either. Additional results by the authors compared DNR and HARP at 95.4% simply because DNR converges at that point, but not further. I think this is an important difference of the algorithm that requires attention.
> > >
> > > These high sparsities around 99.9% is the actual range where HARP becomes essentially beneficial, and I think the authors did a good job in demonstrating that.

---

> > > > ### Comment · Reviewer_j8jC · 2022-11-22
> > > > **Re: On why not agreeing with the authors**
> > > >
> > > > Dear author and reviewer **LsE9**,
> > > >
> > > > I appreciate both of your clarification.
> > > >
> > > > In that regard I would have the following comments:
> > > >
> > > > 1. *"Current state-of-the-art methods for adversarially robust pruning focus on the latter, allocating a
> > > > fixed compression budget governed by the target hardware’s limits and using it uniformly for each
> > > > layer (e.g., Ye et al., 2019; Sehwag et al., 2020)."*
> > > >
> > > > This is added in blue post-rebuttal and definitely needs to be updated. The state-of-the-arts also know how to non-uniformly assign parameters across layers. This sentence makes the state of current art as naïve, which is not.
> > > >
> > > > Authors should note that assuming a paper to baseline (like HYDRA or BSC) is their choice, that does not really draw any boundary to be compared with others if some alternatives are seemingly better than what baseline they have assumed.
> > > >
> > > > 2.  In this line: "entire network to reach a specific target compression $a_t$" .. I am assuming $a_t$ is an user given choice. If so, I am not sure about the argument of the reviewer **LsE9**, on DNR not able to provide user defined compression. DNR, similar to the variable $a_t$, has a target global parameter density $d$ (refer to Algo. 1 of the paper) which is a user choice. So, its still not clear to me or am I missing something here?
> > > >
> > > > 3. The authors have really emphasized on learning different compression ratio at different layers, and that part is not really an additional contribution if the SOTA is updated instead of what are mentioned at the start of Sec. 3. There can be various ways to initialize the layer wise compression, like uniform, ERK, ERK+ etc. It has also been shown such initialization has non-mentionable impact in the final accuracy.
> > > >
> > > > 4. The reviewer has highlighted an important point that major benefits of HARP can be observed at around 99.9% compression. However, if we see table 6 results, the benefits come at the cost of around 24% accuracy drop on clean (compared to DNR/HARP at 95%) and 16% drop on PGD-7 (compared to DNR/HARP at 95%), such huge drop significantly limits the applicability at such high compression. This limits the practical use case of the beneficial region of the proposed method.
> > > >
> > > > 4. Also, can you show the results with DNR at 99% compression? I suppose DNR converges further as the last table with DNR-C has some results with 33.4x compression. So, that requires experimentation before conclusion. Moreover, the trade-off of such algorithms can be adjusted by adjusting the weight of the loss component between clean and adversary. Finally, adding the most competitive results in the Appendix is not a good practice, as again *the baselines are choice of the authors*.
> > > >
> > > > Thus overall, I would request the reviewer **LsE9** to re-asses his/her score in case he/she finds these information useful. I am open to further discussion, as I am open to figure out the merit of this manuscript that may be interesting to the community.
> > > >
> > > > Minor:
> > > >
> > > > 1. Its straight **through** estimator (STE) not straight forward estimator.
> > > > 2. "The exactly formulation of $L_{robust}$.. should be 'exact formulation'..

---

> > > > > ### Comment · Reviewer_LsE9 · 2022-11-23
> > > > > **follow-up**
> > > > >
> > > > > Thanks to Reviewer j8jC for following up! Below are my remarks on these very interesting points.
> > > > >
> > > > > At this point I also want to invite Reviewers phem and GDVf to share opinions and responses to the authors based on their previous comments.
> > > > >
> > > > > (2) I’m not really sure if the density parameter directly matches the final sparsity of the network. Perhaps the objective can be adapted such that one achieves precisely 99.9%. Currently the authors phrase it as: “DNR yields good overall performance but its final compression rate cannot be controlled due to the employed compactness regularization. ”. If this can be in fact manipulated by the authors to perform the experiments suggested at (5), authors should rephrase this.
> > > > >
> > > > > (4) Misunderstanding on “better benefits of HARP observed around 99.9%”: Surely the networks will significantly suffer from huge drops at that level of sparsity. Practically, these sparsity levels may never even be considered as the network weakens a lot, but algorithmically they are worth exploring. Also I believe when it comes to practical use, hardware limitations actually define which level of sparsity one has to establish. At that point I think one may not really have the choice of going with a better performing 95% model. Under hard hardware requirements of x100 model compression for instance, so far HARP looks like a great option.
> > > > >
> > > > > (5) I would like to see this as well if it is experimentally feasible. Of course, as the reviewer pointed out, this may necessitate playing around with the training objective of DNR. However I would add at this point, all methods considered in the paper (HYDRA, BCS etc.) share the same limitation. For instance one can simply better adjust the sampling initialization of BCS such that there are not 2 connections only active at the end, and the algorithm would not perform that poorly. Nevertheless, these are tuning limitations of these previous works.

---

> > > > > > ### Comment · Reviewer_j8jC · 2022-11-23
> > > > > > **Re: follow-up**
> > > > > >
> > > > > > Dear reviewer **LsE9**,
> > > > > >
> > > > > > Here are my thoughts on the follow-up discussion:
> > > > > >
> > > > > > (2) To my understanding and based on the original conference and journal version of DNR (and RigL [1], which is also similar to DNR in terms of keeping compression ratio) can control the compression rate, by the following policy:
> > > > > > a. They first start with the target parameter count, b. every epoch they prune additional p fraction of the weights evenly from each layer, c. they redistribute this additional p-fraction of weights based on layer importance, this policy **ensures that the target compression ratio is controlled** based on user given target. Also, to me their density $d$ is = (1 - $a_t$) for this paper. So, to me both have the ability to adhere to any user given compression ratio.
> > > > > >
> > > > > > (4) Just to clarify, the better performing 95% model has only 5% parameters present, also DNR has shown results up to 33.4x compression, which is essentially around 3% parameters present.
> > > > > >
> > > > > > Having said this, here I and reviewer **LsE9** differ in terms of how much accuracy sacrifice is too much! To me using a 99.5% sparse variant of a large model with 24% clean image sacrifice that too **at small datasets like CIFAR-10**, has significantly limited scope. Most of such use case scenarios would prefer distributed computing instead (based on the literature on hardware and computer architecture) of using a partly unreliable inference model.
> > > > > >
> > > > > > [1] Rigging the Lottery Ticket, ICML 2020.

---

> > > > > > > ### Author Response · Authors · 2022-11-24
> > > > > > > **Reply to further concerns**
> > > > > > >
> > > > > > > Thank you very much for the vivid discussion on our paper. We very much appreciate the feedback.
> > > > > > >
> > > > > > > (2) In DNR, density $d$ indeed equals $1-a_t$, with $a_t$ as used by HARP. However, DNR actually refines the target compression by the pruning rate $p$, offering some "wiggle room" for the final compression rate. HARP, in turn, uses $a_t$ as a fixed target compression. We will clarify this subtle difference in the paper. Thank you!
> > > > > > >
> > > > > > > (4) We thank the reviewer for this alternative viewpoint. Our paper's objective aligns with large portions of related work that push forward to larger compression.
> > > > > > >
> > > > > > > (5a) We would love to show DNR's performance on 99% compression, as we very much agree that such a comparison would be beneficial. Unfortunately, the author's repository only offers code for testing the pruned models in DNR's Table 2, not for training them. Thus, we cannot independently reproduce the results in the paper and cannot extend it to larger models learned on ImageNet, for instance, as offered in our submission. Fig. 3 (ASP-DAC 2021) gives a rough impression of DNR's performance on CIFAR-10 under 99% sparsity. However, the figure does not allow us to derive the exact value reliably. Also, actual values, unfortunately, are not provided in the text. We want to refrain from reciting numbers that we cannot reproduce.
> > > > > > >
> > > > > > > (3)+(5b) We agree with Reviewer j8jC, results on BCS-P should be presented more prominently in the main paper. We are happy to adjust this in the revision. Thus, we have extended our experiments so that BCS-P is initialized with a non-uniform strategy (similar to R-ADMM and HYDRA; cf Section 4.1). This alternative initialization indeed improves BCS-P on 99.9% sparsity, as reported in the table below. Left and right of character ' / ' shows the natural accuracy and PGD-10 robustness. The best and second best approaches are highlighted. Please note that the first (HARP) still outperforms the second best (HYDRA w/ LAMP) significantly.
> > > > > > >
> > > > > > > | Model    | Sparsity | HYDRA         | HYDRA w/ LAMP                           | BCS-P          | BCS-P w/ LAMP                    | HARP                  |
> > > > > > > |----------|----------|---------------|-----------------------------------------|---------------|---------------------------------|-----------------------|
> > > > > > > | ResNet18 | 99%      | 75.53 / 45.84 | **80.16** (+4.63) / **50.07** (+4.32)   | 71.73 / 45.32 | 72.58 (+0.85) / 45.42 (+0.10)   | "**80.25**/**50.36**" |
> > > > > > > |          | 99.9%    | 34.55 / 26.08 | **57.07** (+22.52) / **35.91** (+9.83)  | 21.78 / 12.78 | 56.33 (+34.55) / 35.21 (+22.43) | "**63.99**/**39.39**" |
> > > > > > > | VGG16    | 99%      | 67.33 / 41.47 | **76.75** (+9.42) / **47.96** (+6.49)   | 69.81 / 36.81 | 70.47 (+0.66) / 43.71 (+6.90)   | "**78.58**/**48.71**" |
> > > > > > > |          | 99.9%    | 23.41 / 20.99 | **57.93** (+34.52) / **36.01** (+15.02) | 10.00 / 10.00 | 43.77 (+33.77) / 30.34 (+24.34) | "**59.13**/**37.36**" |
> > > > > > >
> > > > > > > HARP is still able to improve upon "BCS-P non-uniform." Moreover, also HYDRA performs better than BCS-P on 99.9% sparsity, making the work by Sehwag et al. (NeurIPS 2020) a valuable baseline.

---

> > > > > > > > ### Comment · Reviewer_j8jC · 2022-11-25
> > > > > > > > **Re:**
> > > > > > > >
> > > > > > > > I appreciate the authors' effort. Just want to correct them on the comment of prune rate $p$ of DNR or similar methods [1]:
> > > > > > > >
> > > > > > > > **prune rate $p$**: It the rate at which the prune-regrow happens during every epoch for sparse learning, it has nothing to do with the target parameter density $d$, and the target density is always met during every epoch. It is a well known hyper parameter in the **sparse learning** community. Higher prune rate hints higher no of weights will change their locations across layers, while lower hints the opposite. Prune rate of 0 is an extreme case meaning every weights are fixed to their initialized locations, and there will be no prune-regrow to relocate the connections.

---

### Official Review · Reviewer_phem · 2022-10-26

**Confidence:** 4
**Correctness:** 3
**Technical Novelty And Significance:** 2
**Empirical Novelty And Significance:** 3
**Recommendation:** 6

**Clarity, Quality, Novelty And Reproducibility:**

- The design of $\mathcal{L}_{hw}$ in Eq. 2 may not support your explanations below. Considering a case that all layers' expression rate is $a_t$, then the value of $\mathcal{L}_{hw}$ will be $(L - 1)$ which still encourages lower compression rates when L > 1.

- What 's the meaning '$\langle \cdot \rangle$'  in  Eq. 5? The term inside the angle brackets $\in \mathcal{\Theta^{(l)}}$, and $\frac{\part \mathcal{L}}{\part r^{l}}$ and $r$ are scaler, so how do the '$\langle \cdot \rangle$'  map a $\Theta^{(l)}$-dimension vector into a scaler?

**Strength And Weaknesses:**

### Strengths

- The research is promising and practical in the real world. This work meets the need for  compact and adversarial-robust model under safety and computation efficiency critical scenarios.

- Out of the hypothesis that layer-specific compression strategy is equally important as deciding which connections to prune, this work firstly implements non-uniform robust pruning with well-designed optimization process and firm engineering skills.

- A range of experiments are neatly organized to empirically prove the correctness of the starting hypothesis in adversarial robust pruning  and demonstrate the superiority of the proposed method.

### Weaknesses

- There is not much novelty in the fundamental hypothesis in this work. Learnable non-uniform compression strategies have been already have been proven effective in pruning. As more parameters available for optimization, it's intuitively effective for more specific pruning task (e.g., adversarial robust pruning).

- In Sec 4.3, some approaches are tried to explain the superiority of the proposed methods, but it only stucks on listing the found features. The paper could yet be strengthened further by providing more detailed interpretations over the observations.

**Summary Of The Paper:**

This paper proposees a novel method of HARP, which learns both compression rate and mask each layer to achieve a a high reduction rate while maintaining both the clean and adversarial accuracy of original models. Specifically, the method is delicately designed with many skills utilized to implement the non-uniform robust pruning. Experimental results validate the effectiveness of the proposed method.


**Summary Of The Review:**

Excellent implementations to achieve adversarial robust pruning based on insight from previous works. The proposed HARP empirically works with great performance. Since the starting hypothesis is already proposed, more detailed intuitive or theoretical explanations are required to make it distinct with the previous works.

---

> ### Author Response · Authors · 2022-11-14
> **Reply to reviewer phem**
>
> We thank the reviewer for the feedback. We provide answers to the raised questions below and are looking forward to your response to be able to make the appropriate changes to the paper.
>
> ### Ques-1: Lack of novelty in the fundamental hypothesis
>
> The reviewer is correct. Learnable, non-uniform pruning strategies have been explored in prior work, both for traditional pruning (LAMP and ERK) and adversarial robust pruning (BCS-P and MAD). Our method pushes the limits in this line of research improving over novel extensions to HYDRA based on LAMP and ERK, BCS-P as well MAD.
>
> **SKIP** The reviewer only points out that we have no new fundamental hypothesis, which actually is true :/
>
> Moreover, we include two state-of-the-art methods for learnable non-uniform adversarial robust pruning methods **BCS-P** and **MAD** as the comparison objectives. With the comparison in Fig. 1, Fig. 3, Fig. 6, Fig 8 and Table 7, our method presents comparative performance in moderate pruning (90% and 99%) and outperforms in aggressive pruning(>99%). Moreover, Lee et al. [2] presets the outperformed non-uniform strategy **LAMP**, which even achieves better performance than learning based pruning. Thus, we apply **LAMP**'s strategy on Hydra pruning (in Table 6) and the results demonstrate that using non-uniform can improve Hydra pruning. But HARP's setting with concurrently optimizing layer compression rate and neuron conncections presents better effectiveness on the robust pruning. Additionally, as reviewer j8jC suggests, we also offer the comparison with DNR[1] method (cf. Appendix A.1). Our method also shows the comproming performance surpassing DNR's robustness preservation.
>
>
> [1] DNR: A Tunable Robust Pruning Framework Through Dynamic Network Rewiring of DNNs, ASP-DAC 2021.
> [2] Layer-adaptive sparsity for the magnitude-based pruning, ICLR 2021.
>
> ### Ques-2: More detailed interpretation for Sec 4.3
>
> With the strategy visualization in Fig. 4 and further in Fig. 7, we would like to show that Hydra can benefit from strategies determined by LAMP and ERK. HARP yields a similar strategy but preserves more parameters in input and output layers. The results in Table 6 demonstrate the advantage of HARP's behavior.
>
> ### Clarity-1: Loss design in Eq.2
>
> $L_{hw}$  is designed to be zero once we yield the target compression rate. In Eq.2, the $-1$ is supposed to outside the summation of $\frac{1}{a_t} \cdot \sum_{l=1}^{L} \frac{||1_{w \neq \hat{\theta}^{(l)}}||}{\Theta^{(l)}}$. In our revision, we change the Eq.2 as following:
> $L_{hw} = \max [ -1 + \frac{1}{a_t} \cdot \sum_{l=1}^{L} \frac{||1_{w \neq \hat{\theta}^{(l)}}||}{\Theta^{(l)}} , 0 ]$.
>
>
>
> ### Clarify-2: The meaning of $\langle \cdot \rangle$ in Eq.5
>
> The implementation of $\langle \cdot \rangle$ is actually the inner product of the gradient from the objective loss $\mathcal{L}$ to the pruning mask $\mathbf{M}^{(l)}$, which aims to map the gradient matrix to a scalar value [3]. We have added this crucial reference in the revised version of our submission:
>
> [3] Soft Threshold Weight Reparameterization for Learnable Sparsity, ICML 2020.

---

### Author Response · Authors · 2022-11-14
**Reply to all reviewers**

Thanks a lot to all reviewers for your valuable input. We have pro-actively updated our submission to incorporate the reviewer's suggestions. In particular, we:

### Section 1: Introduction.

* Better highlight the made contributions (Reviewer: j8jC Q.3)
* Replace the robustness evaluation with AutoAttack in Fig.1 (Reviewer: LsE9 Q.1)
* Add BCS-P results in Fig.1 (Reviewer: LsE9 Q.3)
* Adopt a logarithmic x-axis in Fig.1 (Reviewer: LsE9 Q.9)

### Section 3: HARP Methodology.

* Clarify that we determine compression (how many parameter to prune) **per layer**. (Reviewer: j8jC Q.4)
* Modify the expression of loss $\mathcal{L}_{hw}$ in Eq.2 and 8 to clarify that `-1` is not part of the sum (Reviewer: phem Q.3)
* Provide an additional reference to derive the gradient matrix mapping to scalar value $r^(l)$ in Eq.5. (Reviewer: phem Q.4)
* Add a caption for the figure in the paragraph on "Balancing pruning objectives" (Reviewer: LsE9 Q.10)

### Section 4: Evaluation
* Include an extended description of the experimental setup in the main part of the paper. (Reviewer: j8jC Q.7, LsE9 Q.2)
* Merge the performance of pre-trained models into Table 1 and 2. (Reviewer: GDVf Q.2)
* Extend the comparison to BCS-P on ResNet18. We present results in Appendix A.2 -> Table 7 and Fig.6 (Reviewer: LsE9 Q.3)
* Add Table 5 showing MAD and HARP side-by-side for a sparsity of 90% and zoom in on the adversarial robustness in Figs.3 and 8. (Reviewer: LsE9 Q.8)
* Visualization BCS-P's strategy in Fig.4 and Fig.7 to analyse performance. (Reviewer: phem Q.2, LsE9 Q.5)

### Appendix:
* Explore the influence of different $\gamma$ step sizes during pruning and summarize results in Appendix A.1. (Reviewer: LsE9 Q.7)
* Add extended comparison with DNR pruning in Appendix A.2. (Reviewer: j8jC Q.1-2)
* Extend comparison to HYDRA's structural pruning in Appendix A.6. (Reviewer: GDVf Q.1)
* Add training consumption in Appendix A.7*. (Reviewer: j8jC Q.6)


We additionally reply to the reviewers' questions in separate comments.

---

### Decision · Program_Chairs · 2023-01-20

**Decision:**

Accept: poster

**Justification For Why Not Higher Score:**

Incremental methodological advance.

**Justification For Why Not Lower Score:**

This paper presents significantly improved robustness performance in various settings, including an extreme case scenario (i.e., 99.9% sparsity) and positive results in a larger scale experiment (i.e., ImageNet). AC also believes that questions and concerns raised by reviewers are mostly addressed in the author’s responses during the rebuttal period, although some reviewers could not make follow-up discussions after their initial review.

**Metareview: Summary, Strengths And Weaknesses:**

This paper presents a new pruning method for adversarially robust neural networks that learns the layer-wise importance scores and compression rate. Specifically, the authors claim that the method balances the two objectives of achieving the adversarial robustness and the compactness of the resultant network by introducing a step-wise regularization weight into the training that dynamically adjusts the focus of the learning process from favoring robustness to achieving the target compression rate. The reviews are divergent, with one accept (LsE9), two marginal accepts (phem, GDVf), and one reject (j8jC). The reasons for acceptance are (i) significant experimental results of the algorithm that well scales to ImageNet, (ii) good compatibility to various adversarial training objectives, and (iii) extensive comparisons among important baselines, including those specifically pointed out by reviewers. The reasons for rejection include (i) incremental methodological advance (e.g., score optimization and layer-wise non-uniform pruning), (ii) requiring a pre-trained model, and (iii) missing comparison to a specific prior work [1].

[1] S. Kundu, M. Nazemi, P. A. Beerel and M. Pedram, “DNR: A Tunable Robust Pruning Framework Through Dynamic Network Rewiring of DNNs,” 2021 26th Asia and South Pacific Design Automation Conference (ASP-DAC), 2021, pp. 344-350.

**Note From Pc:**

if the above contains the word "oral" or "spotlight" please see: "oral" presentation means -> notable-top-5% and "spotlight" means -> notable-top-25%. As stated in our emails, we are disassociating presentation type from AC recommendations